# GRAPHBRIDGE: TOWARDS ARBITRARY TRANSFER LEARNING IN GNNS

**Li Ju, Xingyi Yang, Qi Li, Xinchao Wang**[*]
National University of Singapore
{jujulili,xyang,liqi}@u.nus.edu, xinchao@nus.edu.sg

## ABSTRACT

Graph neural networks (GNNs) are conventionally trained on a per-domain, per-task basis. It creates a significant barrier in transferring the acquired knowledge to different, heterogeneous data setups. This paper introduces **GraphBridge**, a novel framework to enable knowledge transfer across disparate tasks and domains in GNNs, circumventing the need for modifications to task configurations or graph structures. Specifically, GraphBridge allows for the augmentation of any pre-trained GNN with prediction heads and a bridging network that connects the input to the output layer. This architecture not only preserves the intrinsic knowledge of the original model but also supports outputs of arbitrary dimensions. To mitigate the negative transfer problem, GraphBridge merges the source model with a concurrently trained model, thereby reducing the source bias when applied to the target domain. Our method is thoroughly evaluated across diverse transfer learning scenarios, including Graph2Graph, Node2Node, Graph2Node, and graph2pointcloud. Empirical validation, conducted over 16 datasets representative of these scenarios, confirms the framework's capacity for task- and domain-agnostic transfer learning within graph-like data, marking a significant advancement in the field of GNNs. Code is available at https://github.com/jujulili888/GraphBridge.

## 1 INTRODUCTION

With the explosive growth of graph data, the application of Graph Neural Networks (GNNs) has become increasingly widespread in domains (Jing et al., 2022; 2021b; Wu et al., 2020b; Yu et al., 2018; Shah et al., 2020) such as recommendation systems (Gao et al., 2022; Wu et al., 2019) and biopharmaceutics(Zitnik et al., 2018; Rathi et al., 2019). Despite their growing popularity, the effective implementation of GNNs often requires significant training efforts and substantial memory resources. This poses challenges for their practical application in diverse settings. To address these constraints, recent research has focused on reusing pre-trained GNN models (Jing et al., 2023; Yang et al., 2022; Jing et al., 2021a; Deng & Zhang, 2021; Hu et al., 2019a; Sun et al., 2022b; Yang et al., 2020). This approach aims to reduce the need for extensive training, thereby lessening the associated time and resource demands to alleviate the extra training expense.

However, to date, these efforts have not been entirely practical, primarily due to two forms of heterogeneity in graph data. The first is ***task heterogeneity***. The intrinsic non-Euclidean nature of graph data allows for its application across a range of tasks, including graph-level, node-level, and edge-level predictions. However, the graph pre-training paradigm typically assumes consistency between the tasks used in pre-training and those in downstream applications. This becomes problematic when adapting GNNs to new tasks with distinct output formats and knowledge requirements. In such scenarios, GNNs may not perform optimally, as the pre-training might not align well with the demands of these novel tasks.

Beyond task heterogeneity, ***domain heterogeneity*** also poses a significant challenge in transferring knowledge effectively within graph data applications. This refers to the significant differences in node features, connection patterns, and topology across various graph datasets. Consequently, a GNN trained on one specific dataset might struggle to generalize effectively to other datasets with

---

[*]Corresponding Author.

different structures. To mitigate this problem, researchers have been exploring the development of robust GNN transfer frameworks (Hu et al., 2019b; Xia et al., 2022; Zhu et al., 2021b), which aim to enhance the adaptability of GNN models, allowing them to be trained on one graph dataset and then fine-tuned for diverse downstream tasks. However, when the pre-training source domain vastly differs from the downstream task's domain, ***negative transfer*** issues arise, highlighting the need for new approaches to bridge larger domain gaps.

In this paper, we initiate an exploration of a unified workflow for knowledge transfer in diverse graph tasks. Our goal is to overcome the challenges posed by heterogeneity of graphs, especially in the reutilization of knowledge within GNN models. Specifically, we aspire to design a versatile pre-train-tuning graph transfer framework, named "GraphBridge", to facilitate seamless knowledge transfer across diverse graph domains. Rather than updating the parameters of the backbone, we integrate a trainable side network proficient in efficiently guiding end-to-end graph transfer learning.

In realizing our new vision, the primary concern is addressing the challenge of diverse input and output dimensions between source and target tasks. Therefore, we have devised adaptable input dimension adapters, both learnable and non-learnable, tailored to various transfer learning scenarios of differing complexity. Furthermore, we have developed interchangeable prediction heads for different task outputs, including graph classification, node classification, and point cloud classification.

Additionally, to mitigate negative transfer issues that arise when transferring knowledge across distinct domains, we have introduced two effective graph side-tuning techniques called Graph-Scaff-Side-Tune (GSST) and Graph-Merge-Side-Tune (GMST). GSST follows a similar architecture to the ladder-side, while GMST involves the fusion of the pre-trained backbone and a random initialized model with the same architecture, aiming to counteract the negative impact of pre-training knowledge on transfer. Leveraging the advantages of the side-tune branch pathway computations, these modules yield satisfactory results.

To validate the performance of the framework, we have define a "Task Pyramid" for graph transfer learning, as depicted in Figure 1. On 16 graph datasets spanning different tasks, the GraphBridge and the associated Graph Side-tuning approach have been proven effectiveness in different scenarios, even surpassing the performance of full fine-tuning, particularly in challenging tasks.

To conclude, our work makes the following contributions:

• We devised a novel knowledge transfer framework, termed GraphBridge, coupled with an accompanying Graph Side-tuning method to address transfer learning challenges across arbitrary tasks & domains in graph-related applications.

• We have created a "Task Pyramid", which includes four levels of graph transfer tasks across 16 datasets of varying difficulty. We applied our framework to these tasks for comprehensive evaluation.

• Our extended experiments show that our method achieves resource-efficient transfer learning across various task scenarios, pre-training methods, and backbone structures. Remarkably, with only $5\% \sim 20\%$ of the tunable parameter, it delivers comparable performance while consistently handling tasks of different complexities.

## 2 RELATED WORK

**Parameter Efficient Transfer Learning (PETL).** Parameter Efficient Transfer Learning (PETL) is a branch of transfer learning focusing on reducing the computational cost of adapting pre-trained models to new tasks by avoiding updates to the entire parameter set. A popular PETL approach involves delta tuning, which introduces trainable parameters and tuning them. Techniques like adapters(Houlsby et al., 2019; Sung et al., 2022b; Zhang et al., 2021a), LoRA(Hu et al., 2021) and side-tuning(Zhang et al., 2020; Sung et al., 2022a) exemplify this approach. In contrast to adding new parameters to the pre-trained model, prompt-based tuning(Lester et al., 2021; Zhou et al., 2022; Li & Liang, 2021; Li et al., 2023a; 2024) introduces trainable parameters to the input, while keeping the pre-trained model unchanged during training. As GNN pre-training methods(Hu et al., 2019b; Zhu et al., 2021b; Xia et al., 2022; Zhu et al., 2021a; Jin et al., 2020) emerge, the PETL paradigm has gained attention in the GNN domain. Researchers have successfully transferred adapter(Li et al., 2023b) and prompting-based methods(Sun et al., 2023; 2022a) to GNNs. This paper aims to fully

exploit the flexible side-tune structure, designing an efficient graph side-tune method to address severe negative transfer challenges in graph domain tasks.

**Graph Domain Adaptation.** Domain adaptation, a subtopic of transfer learning, seeks to alleviate the negative impact of domain drift in transferring knowledge from a source to a target domain(Pan & Yang, 2009). Particularly prominent in the visual domain, extensive research has been devoted to this area(Ganin et al., 2016; Tan et al., 2017; Long et al., 2015; 2018; Zhuang et al., 2015; Pei et al., 2018). Recent advancements have introduced methods addressing graph domain adaptation tasks, broadly classified into discrepancy-based(Pilancı & Vural, 2020; Vural, 2019), reconstruction-based(Wu et al., 2020a; Cai et al., 2021), and adversarial-based methods(Dai et al., 2022; Shen et al., 2020; Zhang et al., 2019). Despite their efforts to address heterogeneity in various graph knowledge domains for transfer learning, these methods are constrained to scenarios involving tasks at the same level. In contrast, our proposed end-to-end graph transfer learning framework, rooted in the pre-train-finetune paradigm, aims to transcend this constraint, enabling flexible graph transfer learning across diverse domains and tasks.

**Universal Model.** Beyond domain adaptation, it is meaningful for transfer learning to derive a universal model applicable to various downstream tasks, thereby significantly streamlining the process of model pre-training. The exploration of such universal models has been previously conducted in the domains of CV and NLP(McCann et al., 2018; Yu & Huang, 2019; Silver et al., 2021; Reed et al., 2022). Though the field has seen limited engagement, recent endeavors have emerged to develop universal models in non-Euclidean domains(Sun et al., 2023; Jing et al., 2023). In contrast to research on domain adaptation, these models fall short in bridging the substantial domain gaps inherent in different task transfer learning scenarios, leading to the failure to leverage the knowledge of pre-trained models across arbitrary tasks. To tackle the limitation, our framework incorporates the Graph-Merge-Side structure in the tuning stage, which effectively alleviates transfer biases present in the source domain, stemming negative transfer in universal learning.

## 3 METHODS

Achieving transfer learning across arbitrary graph task domains enables GNNs to comprehensively extract general knowledge, laying the foundation for efficient unsupervised graph learning and the development of universal graph model. However, arbitrary domain transfer learning presents challenges that need to be addressed. In this section, we begin by uncovering the key challenges of *Arbitrary Graph Transfer Learning* and outline how we address the challenges posed by various work scenarios I proposed in Figure 1. Particularly, we pinpoint two core problems in our mission: large gap domain adaptation and multi-tasks unification. To handle these challenges, we re-construct the pre-training-tuning framework for graph domains & tasks transfer learning. Additionally, we propose new resource-efficient tuning methods tailored for graphs to mitigate negative transfer.

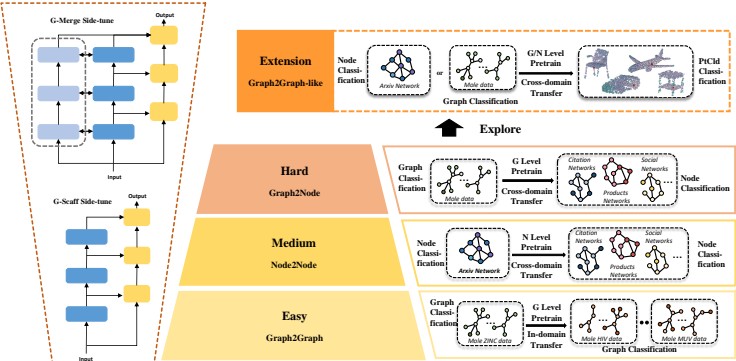

Figure 1: **Task Pyramid & Core Methodology. Left:** Graph side-tuning metods proposed to solve the different difficulty-level of the tasks; **Right:** Graph transfer learning tasks with different levels of difficulty defined in our work.

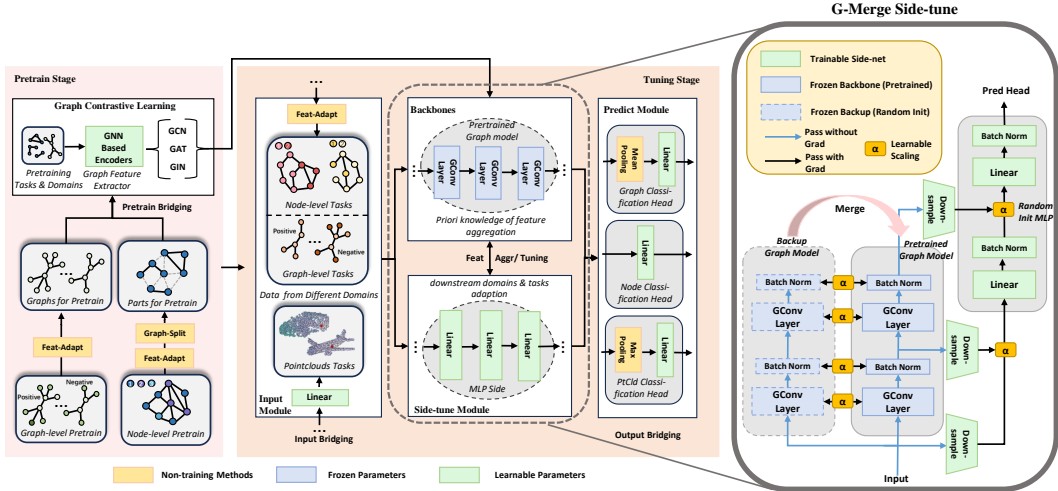

Figure 2: **GraphBridge Framework. Left:** End-to-end GraphBridge framework with 2 stage; **Right:** Architecture of Graph-Merge-Side-Tuning architecture for addressing negative transfer.

## 3.1 CHALLENGES TOWARDS ARBITRARY GRAPH TRANSFER

※ **Multi Input & Output:** The first issue is to fit various input dimensions and output forms of downstream tasks, considering that we only have a frozen pre-trained backbone without any additional dimensional transformation. For instance, the HIV molecular dataset has 8 input features and 1-dimensional output for graph classification, while the Cora network has 1433 input features and output of the same dimension as the number of nodes for node classification. To tackle the multi Input & Output dilemmas, we initially devised an end-to-end transfer learning framework capable of handling arbitrary input-output dimensions.

※ **Domain Gap:** The second challenge towards arbitrary graph transfer lies in the insufficient tuning methods' capacity to make good use of the knowledge from pre-trained models for adaptation in the target domain. In cases where there is a substantial domain gap, the efficacy of full-tuning diminishes, rendering resource-efficient tuning methods even appear as negative transfer. Therefore, we establish innovative graph side-tuning architectures that not only address the issue of negative transfer, but also ensure resource efficiency.

## 3.2 GRAPHBRIDGE FRAMEWORK

In this context, we introduce a pioneering two-stage graph transfer learning framework titled "GraphBridge", which facilitates an end-to-end pre-training-tuning transfer paradigm within the task scenarios outlined in our work. As shown in Figure A.5, our framework comprises a ***Pre-training Stage*** , aimed at extracting generalized graph knowledge, and a ***Tuning Stage*** dedicated to downstream tasks adaptation.

**Pre-training Stage.** In our work, we do not propose new graph pre-training methods. Instead, we design a versatile pre-training stage that can be adapted to various existing graph-level pre-training techniques (since they perform more effectively in graph knowledge learning(Hu et al., 2019b; Sun et al., 2023) in the realm of graph pre-training methods). This adaptability allows our framework to utilize any graph-level pre-training method to obtain the base model for tuning. In our experiments, we employ the effective methods, GraphCL and SimGRACE, for base model pre-training.

**Tuning Stage.** In the tuning stage, our framework comprises three components: Input Bridge, Efficient-tuning, and Output Bridge, which can be tailored to downstream task transfer learning with varying input and output formats.

• **INPUT BRIDGE** adopt the Feat-Adapt methods mentioned above for input feature dimensional adaptation as well. In addition to the non-trainable adapters, we set a trainable linear layer as an adapter specifically for the point cloud dataset.

- **EFFICIENT-TUNING** consists of a pre-trained backbone and tunable side networks for downstream task transfer on adapted inputs. In our work, we design a new graph-side-tuning paradigm which will be discussed in 3.3, instead of using the traditional fine-tuning.

- **OUTPUT BRIDGE** serves as an output adapter for various graph tasks. It integrates several learnable predictor heads tailored to different downstream tasks for graph embeddings obtained from the tuning process. This ensures the generation of appropriate output formats for various graph tasks. For instance, in a graph classification task, a pooling operation followed by a linear prediction head is employed to generate predictions for each graph. In contrast, in a node classification task, a linear head is directly used for predictions, ensuring that each node corresponds to a label.

## 3.3 GRAPH SIDE-TUNING

In the tuning stage, we introduced a novel graph side-tuning technique, enabling effective transfer learning of different graph tasks. On one hand, side-tuning showcases resource efficiency by maintaining performance with fewer parameter manipulations. On the other hand, the flexible architecture of side-tuning facilitates the design of solutions to address negative transfers occurring during large gap domain transfer. For tasks of varied difficulty levels shown in Figure 1, we devised two graph side-tuning methods to tackle the challenges: the elementary *Graph Scaff Side-Tuning* and the advanced *Graph Merge Side-Tuning* methods.

**Graph Scaff Side-Tuning (GSST).** In the design of the Graph Side-tuning, we make an innovation for side network architectures. Unlike the original side-tuning, which used the distillation structure of the base model as the side network, we directly set the side network as a randomly initialised MLP. Such a configuration is pertinent: the high computational time overhead of the graph convolution layer can be circumvented when training the MLP solely using node features. Meanwhile, current research(Han et al., 2022; Zhang et al., 2021b) indicates that MLPs can exhibit graph learning performance comparable to GNNs when guided by the knowledge from GNN models; Hence, the tuning efficiency can be further enhanced while transfer performance being ensured.

The tuning process of GSST is shown in Figure A.5. In our approach, a GNN based model with frozen parameters produces activations at each layer. These activations are then passed through a down-sampling layer and fused layer-wise with the outputs of a trainable MLP side-network. Thus, the loss for downstream task $T_d$ can be formulated as:

$$
\begin{aligned}
\mathcal{L}(\mathbf{x_{T_d}}, \mathbf{y_{T_d}}) = \|\boldsymbol{\alpha_s} \cdot \tilde{f}_{\text{gnn}}(\mathbf{x_{T_d}}, \mathbf{A_{T_d}}; \mathbf{w_g^*}|GNN_{\text{pre}}) \\
+ (\mathbf{1} - \boldsymbol{\alpha_s}) \cdot f_{\text{mlp}}(\mathbf{x_{T_d}}; \mathbf{w_l}) - \mathbf{y_{T_d}} \|
\end{aligned}
\tag{1}
$$

where $\{\mathbf{x_{T_d}}, \mathbf{A_{T_d}}, \mathbf{y_{T_d}}\}$ denotes the node features, adjacency matrix and labels of $T_d$. Moreover, the $\tilde{f}_{\text{gnn}}$ with pre-trained-init parameters $\mathbf{w_g^*}$ and $f_{\text{mlp}}$ with random-init $\mathbf{w_l}$ represent the output of the frozen base model and activated side network respectively, while $\boldsymbol{\alpha_s}$ represents a set of fusion $\alpha$ for each layer between base and side.

As such, Eq. 1 indicates that the proposed GSST fixes the parameters of the base model and adjusts the MLP side network parameters for optimization. Additionally, the alpha blending parameters, as well as the downsampling module for each layer, are updated during back-propagation. The smaller scale of the tuning space reflects the parameter efficiency of our methods. Moreover, by exclusively conducting back propagation on the side network, we avoid the need to compute and retain gradient values for the base model, presenting an additional memory-efficient attribute. At last, the GSST method demonstrates commendable performance in easy task scenarios and the experimental results of it are further elaborated in section 4.2.

**Graph Merge Side-Tuning (GMST).** Nevertheless, in more challenging task scenarios, GSST proves inadequate in bridging the substantial gaps between diverse task domains and knowledge domains. To address the negative transfer problem that occurs when significant domain gap exists, we further propose a novel side-tuning architecture named Graph Merge Side-Tuning (GMST).

In theory, due to the substantial disparity in knowledge between the source domain and the target domain, the base model's involvement tends to trap the model in a local optimum. Therefore, it becomes imperative to mitigate the impact of bias from the source domain on the target domain. Here, we achieve this goal by introducing the backup model to the base-side and fusing it with the original pre-trained model. Specifically, we set up a backup network mirroring the structure of the

pre-trained model, initializing its parameters from random distributions. The parameters of both the backup and the pre-trained model are then frozen, while the parameter merging between each layer of the base model is controlled by the learnable scaling $\alpha$.

During forward stage, the activation of each layer of the base model layer is initially combined by the backup and pre-trained models before being directed to the side network for base-side merging as shown in Figure A.5. Given the two-step model fusion, the back-propagation loss of the algorithm for random downstream task $T_d$ undergoes minor modifications compared to GSST:

$$
\begin{aligned}
\Phi_{\mathrm{b}}(\mathbf{x_{T_d}}) = {} & \boldsymbol{\alpha_b} \cdot \tilde{f}_{\mathrm{gnn1}}(\mathbf{x_{T_d}}, \mathbf{A_{T_d}}; \mathbf{w_g^*}|GNN_{\mathrm{Pre}}) \\
& + (\mathbf{1} - \boldsymbol{\alpha_b}) \cdot \tilde{f}_{\mathrm{gnn2}}(\mathbf{x_{T_d}}, \mathbf{A_{T_d}}; \mathbf{w_g})
\end{aligned}
\tag{2}
$$

$$
\Phi_{\mathrm{s}}(\mathbf{x_{T_d}}) = f_{\mathrm{mlp}}(\mathbf{x_{T_d}}; \mathbf{w_l})
\tag{3}
$$

$$
\mathcal{L}(\mathbf{x_{T_d}}, \mathbf{y_{T_d}}) = \|(\boldsymbol{\alpha_s} \cdot \Phi_{\mathrm{b}}(\mathbf{x_{T_d}}) + (\mathbf{1} - \boldsymbol{\alpha_s}) \cdot \Phi_{\mathrm{s}}(\mathbf{x_{T_d}}) - \mathbf{y_{T_d}}\|
\tag{4}
$$

where $\Phi_{\mathrm{b}}$ and $\Phi_{\mathrm{s}}$ denote the output of base merged activations and side activations, respectively. And $\boldsymbol{\alpha_b}$ refers to the set of fusion parameters for each layer of the base model. $\tilde{f}_{\mathrm{gnn1}}$ and $\tilde{f}_{\mathrm{gnn2}}$ represent pre-trained base model and the random-init. backup model.

As shown in Eq. 2, 3, 4, GMST only adds forwarding of the gradient of the base model fusion parameter to the back-propagation; the gradient of side-tuning itself does not change, which maintains the parameter-efficient attribute of the approach. Meanwhile, Eq. 2 also indicates that by merging the backup and pre-trained model, we can introduce more randomness into the base model, thus diluting the negative impact of the source domain bias on the downstream tasks. Finally, GMST demonstrated significantly better transfer learning than GSST in more difficult task scenarios.

## 4 EXPERIMENTS

We evaluate the performance of our GraphBridge on 16 publicly available benchmarks across four different scenarios defined in Figure 1:

• **Easy**: Transfer learning between graph-level classification tasks within similar knowledge domains, a task frequently explored in existing research on graph pre-training and fine-tuning methods

• **Medium**: Transfer learning between node classification tasks in unrelated knowledge domains.

• **Hard**: Transfer learning between graph classification tasks and node classification tasks in unrelated knowledge domains.

• **Extension**: For extension, we explored the transfer learning between traditional graph data and graph-like (point cloud) data.

As a preliminary study in this field, our goal is not to achieve state-of-the-art performance on all datasets. Instead, we aim to explore the feasibility of a graph-universal model across a diverse range of tasks and datasets.

### 4.1 EXPERIMENTAL SETTINGS

**Datasets.** The datasets employed in our experiments can be categorized based on task levels: graph-level tasks consist of *ZINC-full*, *BACE*, *BBBP*, *ClinTox*, *HIV*, *SIDER*, *Tox21*, *MUV*, *ToxCast*, which is a series of molecular graph datasets; node-level tasks include *ogbn-arxiv*, *Cora*, *CiteSeer*, *PubMed*, *Amazon-Computers*, *Flickr*, encompassing node classification datasets related to citation networks, product ranking networks, and social networks; point cloud tasks involve *ModelNet10*, which is a 10-classification point cloud dataset.

**Model Settings.** In the Graph2Graph task, we employ a five-layer backbone architecture to facilitate the extraction of general knowledge from the extensive ZINC dataset. In the Node2Node, Graph2Node, and Graph2PtCld tasks, we consistently utilize a standard graph neural network structure comprising two-layer graph convolutions. For the backbones of the aforementioned model, we configure the hidden layer dimension of the base to be 100, while the hidden layer dimension of the side network is set to 16.

**Comparison Methods.** In various task scenarios, we employed different comparison methods to assess the performance of GraphBridge. For the evaluation of the Graph2Graph task, we conducted full-stage supervised training, fine- tuning(Zhu et al., 2021b), MetaGPJing et al. (2023), MetaFPJing et al. (2023), and Adapter-GNN(Li et al., 2023b). Given the novelty of the task presented in this paper, there are currently fewer comparison methods available for the last three task scenarios. Therefore, we choose full-stage supervised training, fine-tuning and MetaFP as baselines, and adaptively modify the Adapter-GNN for comparison.

## 4.2 EASY TASK: GRAPH2GRAPH TRANSFER

For the easy-level task, we chose the largest-scale dataset, ZINC-full, as the pre-training dataset, and utilize the remaining molecular datasets for transfer learning. Additionally, as the Adapter algorithm is specifically designed for the GIN model, we exclusively used GIN as the backbone of the base model in our experiments for a fair comparison. As depicted in Table 1, our GSST method has demonstrated robust performance in the Graph2Graph task, outperforming the fine-tuning method by 0.6% and 0.1% under different pre-training approaches and significantly working better compared to other efficient tuning methods. Examining the errors, it is evident that the error fluctuations of the GSST algorithm are consistently below 1%, highlighting its convergence stability. Moreover, a comparison of the last column indicates that our algorithm exhibits enhanced robustness compared to baselines, consistently delivering performance improvements across different pre-training methods.

Table 1: **Results of Graph2Graph Transfer.** : Test ROC-AUC (%) performances on molecular prediction benchmarks with different pre-train-tuning workflows. **Imp.** refers to the improvement of parameter-efficient tuning methods in comparison to the fine-tuning.

| Pre-train Methods | Tuning Methods | BACE | BBBP | ClinTox | HIV | SIDER | Tox21 | MUV | ToxCast | Avg. | Imp. |
|---|---|---|---|---|---|---|---|---|---|---|---|
| GraphCL | FT | $74.6_{\pm2.2}$ | $68.6_{\pm2.3}$ | $69.8_{\pm2.2}$ | $78.5_{\pm1.2}$ | $59.6_{\pm0.7}$ | $74.4_{\pm0.5}$ | $73.7_{\pm2.7}$ | $62.9_{\pm0.4}$ | 70.3 | – |
| | MetaGP | $72.5_{\pm1.1}$ | $66.9_{\pm1.4}$ | $67.7_{\pm2.5}$ | $77.3_{\pm2.2}$ | $59.0_{\pm1.8}$ | $72.5_{\pm1.4}$ | $74.4_{\pm3.0}$ | $62.2_{\pm0.4}$ | 69.1 | -1.2% |
| | MetaFP | $75.3_{\pm3.6}$ | $66.4_{\pm2.1}$ | $70.3_{\pm1.2}$ | $75.6_{\pm1.3}$ | $59.2_{\pm3.3}$ | $74.4_{\pm0.2}$ | $74.8_{\pm2.8}$ | $63.0_{\pm2.3}$ | 69.9 | -0.4% |
| | Adapter | $76.1_{\pm2.2}$ | $67.8_{\pm1.4}$ | $72.0_{\pm3.8}$ | $77.8_{\pm1.3}$ | $59.6_{\pm1.3}$ | $74.9_{\pm0.9}$ | $75.0_{\pm2.1}$ | $63.1_{\pm0.4}$ | 70.7 | 0.4% |
| | **GSST** | $79.3_{\pm0.2}$ | $69.5_{\pm1.0}$ | $71.1_{\pm0.4}$ | $72.8_{\pm0.9}$ | $60.6_{\pm0.1}$ | $72.1_{\pm0.1}$ | $78.0_{\pm0.7}$ | $62.9_{\pm0.1}$ | **70.9** | 0.6% |
| SimGRACE | FT | $74.7_{\pm1.0}$ | $65.5_{\pm1.0}$ | $53.8_{\pm2.3}$ | $74.6_{\pm1.2}$ | $58.1_{\pm0.6}$ | $71.9_{\pm0.4}$ | $71.0_{\pm1.9}$ | $61.3_{\pm0.4}$ | 66.3 | – |
| | MetaGP | $72.2_{\pm3.1}$ | $59.8_{\pm1.8}$ | $49.6_{\pm2.5}$ | $69.6_{\pm1.3}$ | $57.7_{\pm2.0}$ | $70.7_{\pm1.7}$ | $71.2_{\pm2.1}$ | $61.6_{\pm2.4}$ | 64.3 | -2.0% |
| | MetaFP | $74.0_{\pm2.3}$ | $62.2_{\pm2.1}$ | $52.3_{\pm3.0}$ | $70.3_{\pm2.6}$ | $58.2_{\pm3.5}$ | $71.9_{\pm1.8}$ | $72.8_{\pm2.7}$ | $61.1_{\pm1.9}$ | 65.4 | -0.9% |
| | Adapter | $74.9_{\pm1.7}$ | $64.6_{\pm1.3}$ | $53.9_{\pm2.0}$ | $72.3_{\pm1.2}$ | $57.2_{\pm0.9}$ | $71.4_{\pm0.6}$ | $71.8_{\pm1.4}$ | $61.3_{\pm0.6}$ | 65.9 | -0.4% |
| | **GSST** | $73.0_{\pm0.6}$ | $65.4_{\pm0.2}$ | $57.2_{\pm0.3}$ | $69.1_{\pm0.1}$ | $57.9_{\pm0.2}$ | $72.3_{\pm0.3}$ | $74.4_{\pm0.5}$ | $61.6_{\pm0.1}$ | **66.4** | 0.1% |

## 4.3 MEDIUM TASK: NODE2NODE TRANSFER

In the Node2Node transfer scenario, ogbn-arxiv was selected for model pre-training, while the remaining node classification datasets were used for validation.

We show in Table 2 the results of the Graph2Graph transfer task. The 7th and 12th lines of Table 2 exhibit that GMST demonstrated superior transfer learning performance across most datasets when compared to the baselines. This was particularly notable in the GIN backbone settings of PubMed and CiteSeer, where GMST outperformed training from scratch by 6.8% and 3.7%, respectively. The proposed method consistently maintains stable performance across various pre-training methods and GNN backbones, showcasing the universality of GraphBridge. Although, on the Flickr, Amazon datasets, the performance of the proposed GMST method is slightly inferior to that of the fine-tune method, it still outperforms other efficient tuning methods. This suggests that the parameter-efficient GMST method may not exhibit significant advantages when the task domain's scope is not expansive enough, but achives SOTA among the efficient tuning methods.

## 4.4 HARD TASK: GRAPH2NODE TRANSFER

For the Graph2Node transfer scenario, we opted for a relatively larger HIV dataset for models pre-training, while using the same validation dataset as in the Node2node for transfer learning.

Table 2: **Results of Node2Node Transfer.** Test Acc. (%) on diverse node-classification benchmarks with different tuning methods under node-level data pre-training. We conducted experiments with two pre-training methods and three GNN backbones. *Due to space constraints, the error bars for the experiments are shown in Figure 9 .

| Pre-train Methods | Tuning Methods | Citeseer | | | PubMed | | | Cora | | | Amazon | | | Flickr | | |
|---|---|---|---|---|---|---|---|---|---|---|---|---|---|---|---|---|
| | | GCN | GAT | GIN | GCN | GAT | GIN | GCN | GAT | GIN | GCN | GAT | GIN | GCN | GAT | GIN |
| — | Scratch Train | 64.30 | 69.21 | 55.10 | 75.70 | 75.10 | 65.80 | 76.90 | 77.00 | 72.10 | 92.37 | 92.33 | 91.89 | 53.07 | 52.97 | 53.15 |
| GraphCL | FT | 56.60 | 56.80 | 52.80 | 69.90 | 70.20 | 67.30 | **74.40** | **73.30** | 62.40 | **92.22** | **92.00** | **91.02** | **53.32** | **52.85** | **53.90** |
| | MetaFP | 53.50 | 55.20 | 54.50 | 65.40 | 68.10 | 65.20 | 65.40 | 67.10 | 60.80 | 86.67 | 87.27 | 82.36 | 45.52 | 45.49 | 44.37 |
| | Adapter | - | - | 55.20 | - | - | 65.40 | - | - | 62.40 | - | - | 85.28 | - | - | 50.21 |
| | **GSST** | 54.00 | 55.80 | 56.40 | 69.80 | 71.80 | 69.00 | 63.30 | 64.00 | 59.10 | 88.95 | 84.77 | 85.13 | 49.72 | 44.32 | 49.54 |
| | **GMST** | **59.30** | **63.40** | **58.80** | **72.10** | **75.00** | **72.60** | 73.10 | 72.30 | **65.40** | 89.42 | 90.19 | 86.15 | 51.92 | 47.70 | 49.94 |
| SimGRACE | FT | 58.90 | 57.60 | 45.50 | 71.30 | 71.70 | 64.10 | 72.90 | 71.20 | 64.40 | **92.37** | **92.29** | **91.28** | **53.60** | 50.81 | **53.77** |
| | MetaFP | 54.20 | 55.30 | 46.60 | 67.20 | 68.50 | 65.70 | 66.30 | 63.40 | 60.20 | 83.45 | 85.42 | 80.54 | 47.74 | 43.56 | 48.75 |
| | Adapter | - | - | 48.40 | - | - | 63.20 | - | - | 61.80 | - | - | 80.22 | - | - | 51.23 |
| | **GSST** | 52.00 | 52.10 | 49.50 | 68.00 | 70.00 | 67.30 | 64.60 | 59.30 | 53.90 | 88.84 | 87.86 | 80.26 | 48.93 | 45.20 | 49.71 |
| | **GMST** | **61.60** | **63.40** | **58.90** | **73.20** | **75.80** | **72.70** | **75.10** | **72.20** | **66.70** | 90.88 | 90.53 | 84.19 | 50.56 | 47.71 | 51.16 |

As can be seen in Table 3, the merits of the GMST become more pronounced in more challenging task: For the Cora dataset, GMST consistently outperforms fine-tuning by 5-10% across different pre-training methods and backbones. Although on the Amazon and Flickr datasets, GMST does not strictly surpass the performance of fine-tuning, the gap between the results of the two tuning methods narrows. Besides, its advantages over other efficient tuning methods extend further. A comparison between Table 3 and 2 reveals that, for identical downstream tasks, all other tuning methods experience significant performance degradation, whereas our GMST method keeps a competitive edge. This result aligns with expectations: with the expansion of the domain, the knowledge extracted from the original domain tends to have a negative impact for the downstream task. In contrast, the backup module introduced by the GMST mitigates this influence, accelerating model convergence on downstream tasks.

Table 3: **Results of Graph2Node Transfer.** Test Acc. (%) on diverse node-classification benchmarks with different tuning methods under graph-level data pre-training. We conducted experiments with two pre-training methods and three GNN backbones. *Due to space constraints, the error bars for the experiments are shown in Figure 10 .

| Pre-train Methods | Tuning Methods | Citeseer | | | PubMed | | | Cora | | | Amazon | | | Flickr | | |
|---|---|---|---|---|---|---|---|---|---|---|---|---|---|---|---|---|
| | | GCN | GAT | GIN | GCN | GAT | GIN | GCN | GAT | GIN | GCN | GAT | GIN | GCN | GAT | GIN |
| — | Scratch Train | 64.30 | 69.21 | 55.10 | 75.70 | 75.10 | 65.80 | 76.90 | 77.00 | 72.10 | 92.37 | 92.33 | 91.89 | 53.07 | 52.97 | 53.15 |
| GraphCL | FT | 52.60 | 49.90 | 46.90 | 68.60 | 68.00 | 63.50 | 69.20 | 60.90 | 63.10 | **91.82** | 89.09 | **90.44** | **52.34** | **49.87** | **52.38** |
| | MetaFP | 50.40 | 49.80 | 45.50 | 65.90 | 66.30 | 60.40 | 64.30 | 61.00 | 60.30 | 85.54 | 86.11 | 80.38 | 48.42 | 44.79 | 42.63 |
| | Adapter | - | - | 46.10 | - | - | 59.40 | - | - | 57.80 | - | - | 82.77 | - | - | 48.67 |
| | **GSST** | 48.70 | 49.90 | 50.20 | 64.60 | 64.40 | 65.80 | 52.20 | 51.30 | 56.00 | 83.68 | 80.77 | 80.23 | 48.22 | 44.49 | 47.55 |
| | **GMST** | **61.90** | **62.40** | **57.90** | **73.10** | **73.70** | **73.90** | **74.80** | **72.30** | **66.50** | 88.62 | **89.79** | 85.50 | 51.30 | 47.34 | 49.54 |
| SimGRACE | FT | 53.70 | 53.00 | 43.30 | 59.30 | 68.10 | 61.80 | 65.00 | 64.30 | 57.70 | **92.26** | 89.46 | **90.88** | **52.42** | 48.05 | **53.50** |
| | MetaFP | 51.30 | 52.30 | 44.10 | 56.40 | 61.40 | 56.30 | 61.10 | 64.50 | 56.80 | 85.24 | 86.49 | 81.76 | 46.58 | 46.99 | 44.37 |
| | Adapter | - | - | 45.40 | - | - | 59.90 | - | - | 56.60 | - | - | 80.43 | - | - | 46.32 |
| | **GSST** | 49.90 | 45.40 | 51.90 | 58.20 | 61.60 | 65.30 | 54.50 | 49.20 | 48.30 | 80.95 | 81.68 | 77.39 | 47.80 | 44.86 | 48.93 |
| | **GMST** | **63.00** | **62.20** | **58.50** | **72.70** | **74.80** | **73.30** | **74.30** | **71.10** | **65.20** | 89.67 | **89.46** | 85.26 | 50.24 | 47.29 | 51.57 |

## 4.5 EXTENSION TASK: GRAPH2PTCLD TRANSFER

Here, the pre-trained model acquired from Node2Node and Graph2Node tasks served as the base models backbone for the Graph2PtCld task. Subsequently, we transferred the model to adapt to the ModelNet10 task. In this scenario, we seek to explore whether the GraphBridge framework for graph tasks can transfer knowledge learned from graph domains to graph-like data.

The experimental results presented in Table 4 affirm the capability of our proposed GraphBridge framework for achieving transfer learning from graphs to those graph-like data, since our methods demonstrates significant performance improvements against previous works. According to the

extensive exploration, we were surprised to observe that the GSST method demonstrated superior performance compared to the GMST method in the final results when tuning with a pre-trained backbone on graph-level datasets. It even outperformed all other tuning methods, including fine-tuning. This phenomenon can be explained by considering the dataset: ModelNet10 is a 10-classified point cloud dataset, which exhibits organizational similarities to a graph classification task. Therefore, using the molhiv pre-training results as the backbone of the model for transfer learning towards the point cloud classification task can be considered a Graph2Graph transfer task, a scenario where GSST excels. However, when employing the arxiv dataset for backbone pre-training, the situation changes. In contrast to the consistent excellence exhibited by GSST in scenarios where the original task is graph classification, GMST maintains stability in the face of a larger domain gap.

Table 4: **Results of Graph2PtCld Transfer.** We configure pre-trained models on graph-level and node-level data across distinct types of graph layers as the initialization for backbone.

| Pre-train Methods | Tuning Methods | Graph-level: ogbg-molhiv | | | Node-level: ogbn-arxiv | | |
|---|---|---|---|---|---|---|---|
| Backbones | | GCN | GAT | GIN | GCN | GAT | GIN |
| — | Scratch Train | $77.4_{\pm1.8}$ | $81.6_{\pm1.1}$ | $87.9_{\pm1.0}$ | $77.4_{\pm2.1}$ | $81.6_{\pm1.5}$ | $87.9_{\pm1.3}$ |
| GraphCL | FT | $78.1_{\pm1.7}$ | $74.9_{\pm2.1}$ | $83.9_{\pm1.1}$ | $73.4_{\pm3.0}$ | $\mathbf{75.4_{\pm2.4}}$ | $\mathbf{87.3_{\pm3.1}}$ |
| | MetaFP | $73.1_{\pm3.6}$ | $70.3_{\pm2.4}$ | $78.8_{\pm1.6}$ | $70.2_{\pm2.5}$ | $71.9_{\pm2.8}$ | $74.1_{\pm2.0}$ |
| | Adapter | - | - | $80.9_{\pm2.2}$ | - | - | $75.9_{\pm2.8}$ |
| | GSST | $\mathbf{81.7_{\pm2.9}}$ | $\mathbf{78.6_{\pm3.7}}$ | $\mathbf{84.7_{\pm2.8}}$ | $70.4_{\pm3.1}$ | $74.0_{\pm2.7}$ | $80.4_{\pm3.8}$ |
| | GMST | $77.8_{\pm2.6}$ | $74.8_{\pm3.5}$ | $82.5_{\pm2.4}$ | $\mathbf{75.6_{\pm3.4}}$ | $74.5_{\pm2.7}$ | $80.8_{\pm2.7}$ |
| SimGRACE | FT | $78.6_{\pm1.5}$ | $70.9_{\pm1.8}$ | $77.3_{\pm1.7}$ | $72.8_{\pm2.0}$ | $78.0_{\pm2.3}$ | $\mathbf{79.7_{\pm1.9}}$ |
| | MetaFP | $71.7_{\pm1.2}$ | $66.9_{\pm2.2}$ | $76.5_{\pm1.5}$ | $69.8_{\pm2.3}$ | $71.0_{\pm1.7}$ | $74.4_{\pm2.0}$ |
| | Adapter | - | - | $76.9_{\pm3.2}$ | - | - | $73.8_{\pm2.9}$ |
| | GSST | $\mathbf{79.3_{\pm4.2}}$ | $\mathbf{81.8_{\pm3.7}}$ | $\mathbf{77.3_{\pm2.3}}$ | $71.8_{\pm3.8}$ | $69.2_{\pm2.8}$ | $63.9_{\pm2.0}$ |
| | GMST | $65.2_{\pm2.7}$ | $65.1_{\pm3.1}$ | $76.4_{\pm2.7}$ | $\mathbf{78.1_{\pm3.6}}$ | $\mathbf{78.1_{\pm3.0}}$ | $78.7_{\pm3.2}$ |

In summary, GraphBridge demonstrates convincing performance across different task scenarios. While the proposed framework does not surpass full fine-tuning in all experimental settings, it consistently outperforms the previous efficient tuning methods. Based on the experimental results, we also outline the specific application scenarios for GSST and GMST. Specifically, GSST is most suitable for scenarios where the domain gap between the pre-training task and the downstream task is minimal, e.g. Graph2Graph and Graph2PtCld; Conversely, GMST is more effective when there is a significant gap between source and target domains, e.g. Node2Node, Graph2Node, and Node2PtCld. Furthermore, hard tasks encompass not only the Graph2Node displayed in the main paper, but also other transfer scenarios such as Node2Graph, Graph2Edge and so on. For experiments of supplementary hard scenarios, please refer to Appendix A.4.

## 4.6 ABLATION STUDIES

**Tuning Efficacy.** To evaluate the efficacy of the proposed GSST and GMST for arbitrary graph transfer, we assess the tuning outcomes across varying levels of pre-training bases and diverse backbones, using the Cora dataset as a representative. As illustrated in Table 5, using the pre-trained model directly for inference in the downstream task without fine-tuning results in unacceptable performance. Our proposed parameter-efficient tuning method effectively transfers the pre-trained model to the downstream task. As the transfer task transitions from moderate to difficult, the performance of GSST gradually declines. In contrast, GMST exhibits sustained effectiveness, showcasing its robust capability in mitigating the challenges associated with negative transfer.

**Resource Efficiency.** In the discussion of efficiency of our tuning methods, We validate from two distinct perspectives: parameter efficiency and tuning efficiency.

In terms of parameter efficiency, Figure 3 illustrates the adjustable parameters of different tuning methods across various backbone architectures. Since MetaGP and MetaFP are prompt-tuning methods, their tunable parameters are determined by the datasets, rather than the GNN backbone architecture. Therefore, the tunable parameters scale of both MetaGP and MetaFP are averaged across various datasets, keeping same for different backbones. Notably, our GSST and GMST ex-

hibit significantly fewer tuning parameters compared to most alternative methods, especially in the GIN backbone, where we have only 5% of their parameters. Additionally, due to the fixed MLP structure in our side network, the scale of our parameter space remains constant across diverse pre-training backbones, which has the same advantage as the model-free prompt tuning approaches.

Table 5: **Ablation studies on the effects of GSST and GMST modules designed for arbitrary graph transfer learning.** we conduct tests under three training scenarios: direct inference without tuning, tuning with GSST, and tuning with GMST. The results are presented in terms of test Acc. (%) under various backbones pre-trained by GraphCL strategy.

| | | Dataset: **Cora** | | | | | |
|---|---|---|---|---|---|---|---|
| GSST | GMST | Node-level Pretrain | | | Graph-level Pretrain | | |
| | | GCN | GAT | GIN | GCN | GAT | GIN |
| ✗ | ✗ | 13.80 | 11.60 | 17.30 | 13.50 | 31.90 | 17.00 |
| ✓ | ✗ | 63.30 | 64.00 | 59.13 | 52.20 | 51.30 | 56.00 |
| ✗ | ✓ | **73.10** | **72.32** | **65.40** | **74.80** | **72.30** | **66.50** |

As for tuning efficiency, we validate the speed-up of different tuning methods compared to scratch training on the Cora dataset. Here, we computed the relative speed-up by measuring the convergence time for each tuning method. Table 6 presents the results, demonstrating that our proposed transfer learning method significantly accelerates the scratch training process. Furthermore, when compared to other tuning techniques, both GMST and GSST exhibit substantially higher speed-up rates than fine-tuning and AdapterGNN. The comparatively lower speed-up of our method relative to MetaFP can be attributed to the model-free characteristic of MetaFP, since the gradient update in data is faster than the process in model parameter. Overall, the validation of training speed-up indicates that our method delivers high performance while maintaining strong efficiency.

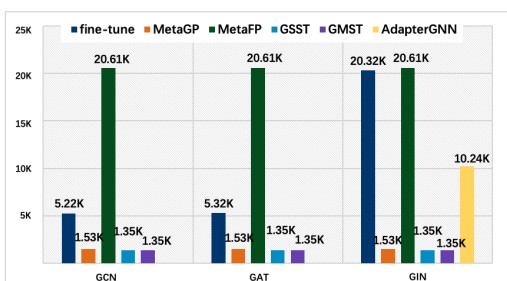

| Speed-up. $\uparrow$ (%) | GCN | GAT | GIN |
|---|---|---|---|
| FT | 3.3 | 7.2 | 1.7 |
| MetaFP | 77.3 | 68.6 | 74.3 |
| Adapter | - | - | 20.2 |
| **GSST** | 39.4 | 57.6 | 26.3 |
| **GSMT** | 31.6 | 52.3 | 20.8 |

Figure 3: **Adjustable parameter sizes in different tuning algorithms across distinct backbones.** We conduct statistics on five-layer backbones.

Table 6: **Training speed-up of different tuning methods compared to Scratch Training.** We selected the transfer experiment on Cora dataset in the challenging scenario as a representative.

The validation conducted demonstrates the effectiveness of our approach in seamlessly transferring the commonly used pre-train + tuning framework from CV and NLP to graph domain research. This transfer is substantiated from both efficacy and efficiency standpoints, marking an initial success in achieving seamless transfer learning across diverse tasks in the graph domain.

## 5 CONCLUSIONS

In this paper, we introduce a novel GraphBridge framework for resource-efficient graph transfer learning toward arbitrary downstream tasks and domains. Our goal is to create a unified workflow that maximizes the utility of pre-trained Graph Neural Networks (GNNs) for various cross-level and cross-domain downstream tasks, eliminating the need for data reorganization and task reformulation. To this end, we have established four scenarios for graph transfer learning tasks, ranging from easy to complex, and proposed two resource-efficient tuning methods, namely GSST and GMST, to resolve the dilemmas. Our experiments, conducted on selected datasets across different domains and tasks—including graph and node classification, as well as 3D object recognition—demonstrate the effectiveness of our approach in achieving arbitrary domain transfer learning on GNNs with improved resource efficiency. Nevertheless, there are still constraints in our experimental setup. In our future work, we will strive to tackle transfer tasks across more benchmarks using our GraphBridge.

ACKNOWLEDGEMENT

This project is supported by the National Research Foundation, Singapore, under its AI Singapore Programme (AISG Award No: AISG2-RP-2021-023).

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

# A    APPENDIX

This document provides an in-depth analysis of our proposed methodology, offering additional insights and experimental details that complement our main findings and enhancing the understanding of our methods. Specifically, in Section A.1, we delve into the intricacies of various datasets utilized in our research, shedding light on their unique characteristics and relevance to the study. Section A.2 and Section A.3 is dedicated to a comprehensive ablation study, where we critically evaluate different architectural configurations and their impact on the performance of our proposed models. Finally, we further investigated the performance of GraphBridge in supplemental hard transfer scenarios in A.4, including Node2Graph and Graph2Edge, to guarantee its generalizability.

## A.1    DATASETS DETAILS

We provide in Table 7 the statistics of several graph benchmarks used in the main manuscript. This section aims to highlight the diversity and range of our benchmark datasets, showcasing their varied characteristics and applications.

Table 7: **Summary of the 16 datasets used in the main manuscript and supplementary material.**

| Names | Task Descriptions | Feature Dimensions | Nodes | Edges | # Graphs |
|---|---|---|---|---|---|
| 1. Flickr | Online Images Classification | 500 | 89,250 | 899,756 | 1 |
| 2. Cora | Machine-Learning Paper Classification | 1,433 | 2,708 | 5,429 | 1 |
| 3. Citeseer | Computer-Science Paper Classification | 3,703 | 3,327 | 4,732 | 1 |
| 4. Pubmed | Diabete-related Publication Classification | 500 | 19,717 | 44,338 | 1 |
| 5. ogbn-arxiv | Subject Area Prediction of arXiv Papers | 128 | 169,343 | 1,166,243 | 1 |
| 6. Amazon Computers | Computer-Product Category Prediction | 767 | 13,752 | 574,418 | 1 |
| 7. BACE | Molecule Property Classification | 2 | $\sim 34.1$ | $\sim 73.7$ | 1,513 |
| 8. BBBP | Molecule Property Classification | 2 | $\sim 23.9$ | $\sim 51.6$ | 2,039 |
| 9. ClinTox | Molecule Property Classification | 2 | $\sim 26.1$ | $\sim 55.5$ | 1,484 |
| 10. HIV | Molecule Property Classification | 2 | $\sim 25.5$ | $\sim 54.9$ | 41,127 |
| 11. SIDER | Molecule Property Classification | 2 | $\sim 33.6$ | $\sim 70.7$ | 1,427 |
| 12. Tox21 | Molecule Property Classification | 2 | $\sim 18.6$ | $\sim 38.6$ | 7,831 |
| 13. MUV | Molecule Property Classification | 2 | $\sim 24.2$ | $\sim 52.6$ | 93,087 |
| 14. ToxCast | Molecule Property Classification | 2 | $\sim 18.7$ | $\sim 38.4$ | 8,597 |
| 15. ZINC-full | Molecule Property Classification | 2 | $\sim 23.2$ | $\sim 49.8$ | 249,456 |
| 16. ModelNet10 | 3D Object Recognition | 3 | $\sim 9,508.2$ | $\sim 37,450.5$ | 4,899 |

**Image Relation Dataset.** Specifically, the Flickr originates from NUS-wide (Zeng et al., 2019) which contains 89,250 nodes and 899,756 edges. One node in the graph represents one image uploaded to Flickr. If two images share some common properties (e.g., same geographic location, same gallery, comments by the same user, etc.), there is an edge between the nodes of these two images. Node features are the 500-dimensional bag-of-word representation of the images provided by NUS-wide. For labels, each image belongs to one of the 7 classes.

**Citation Network Dataset.** The following three datasets, i.e., Cora, Citeseer and Pubmed (Sen et al., 2008), are all citation network datasets used for single-label node classification. Specifically, both Cora and Citeseer contain publications on computer science. Pubmed, on the other hand, only comprises the papers pertaining to diabetes. Moreover, ogbn-arxiv dataset(Wang et al., 2020; Hu et al., 2020a) contains a directed graph, which denotes the citation network among all Computer Science (CS) papers in arXiv, with each node representing an arXiv paper and each directed edge indicating that one paper cites another one. The node features are the average embeddings of words in their title and abstract, which are computed by using the skip-gram model.

**Good Purchase Dataset.** Amazon Computers and Amazon Photo are the segments of the Amazon co-purchase graphs from (McAuley et al., 2015), where the nodes represent various goods, labeled by the corresponding product categories. Here we chose Amazon-Computers with more samples for our validation.

**Molecular Structure Dataset.** BACE, BBBP, ClinTox, HIV, SIDER, Tox21, MUV, ToxCast and ZIN-full are all molecular property prediction datasets proposed by (Wu et al., 2018). Every graph in these datasets denotes a molecule, with the nodes representing atoms, and edges denoting the chemical bonds. The node features contain the atomic number and chirality and the additional atom features, e.g., the formal charge. The number of prediction tasks varies across molecular datasets, and each task corresponds to a binary classification for molecular properties

**Point Cloud Dataset.** For the task of point cloud classification, we adopt the `ModelNet10` dataset (Wu et al., 2015), which is a subset of ModelNet40. Here, we chose ModelNet10 with fewer samples to speed up our experiments. Specifically, the `ModelNet10` dataset contains 4899 CAD models of 10 man-made object categories, of which 3991 CAD models (ModelNet40: 40 classes-classification dataset with 9,843 CAD models are used for training and 2,468 CAD models are for testing) are used for training and 908 CAD models are for testing. For each CAD model, we sample 1,024 3D points from the mesh surfaces and also rescale the associated coordinates to fit into the unit sphere.

## A.2 ABLATION FOR NETWORK ARCHITECTURE

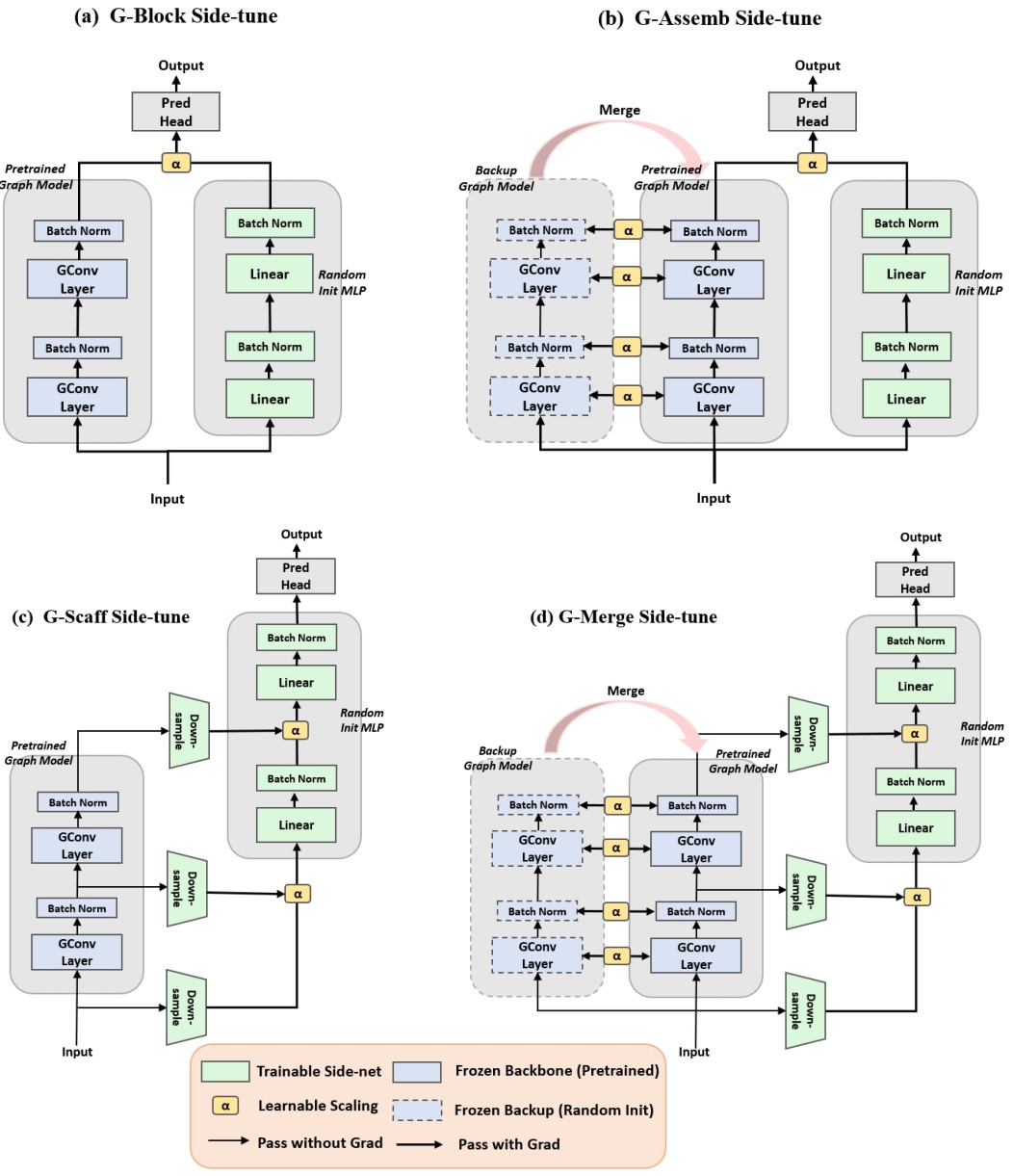

Figure 4: **All versions of Graph Side-tuning architectures. (a) G-Block Side-tune:** The Simplest version of the Graph Side-tuning architecture, which has separated base and side networks; **(b) G-Assemb Side-tune:** G-Block Side-tune with a backup model designed in base model for negative transfer alleviation; **(c) G-Scaff Side-tune:** Graph Side-tune architecture with layer-wise fusion between base model and side network; **(d) G-Merge Side-tune:** G-Scaff Side-tune with a backup model designed in base for negative transfer alleviation.

Before proposing Graph Scaff Side-Tuning (GSST) and Graph Merge Side-Tuning (GMST), we first designed a simple version of the Graph Side-Tuning structures based on (Zhang et al., 2020), known as the Graph Block Side-Tuning (GBST) and Graph Assemb Side-Tuning (GAST), respectively as shown in Figure 4. In this section, our aim is to demonstrate the superiority of GSST and GMST over these simpler predecessors, GBST and GAST, by comparing their respective performances and effectiveness in various applications.

**Architecture Setup.** GBST, a simplified version of GSST, differs slightly in the fusion of the base model and side network compared to GSST. In contrast to the layer-wise fusion approach of GSST, GBST keeps the base model and side network independent until the final output, where they are fused. Similarly, as a simplified version of GMST, GAST differs from GMST only in the side-network fusion phase; however, the fusion of the backup and pre-trained models follows the same layer-by-layer fusion paradigm exactly.

However, such straightforward architectures do not yield satisfactory performance in arbitrary graph transfer learning. This deficiency arises because, without a layer-wise connection between the side network and the base network, the side network loses crucial layer-specific information during the training process. Graph convolution operates uniquely at each layer, contributing to distinct information aggregation functions in the forward propagation of the graph network. The side network, represented by an MLP, needs to learn these layer-specific information aggregation functions independently, as it lacks the structured graph convolution inherent in the original graph. This is essential for achieving a comparable generalization ability in addressing graph-related problems.

While these two simple Graph Side-tuning modules may not effectively address our problem, they have provided valuable insights that guided the formulation of our final architectures. Following the same task scenario outlined in the main text, we extended our experiments to include GBST and GAST across four levels of difficulty: easy, medium, difficult, and extension. The results, along with those of other methods, are presented in the subsequent tables. Furthermore, we evaluated the results of GBST and GAST collectively in all supplementary experiments presented in Appendix.

**Results.** As evident from Tables 8, 9, 10, the simplified version of Graph Side-tuning performs less effectively than the final version across tasks of varying difficulty levels—easy, medium, and hard. Specifically, on hard tasks, the results of the GBST and GAST could not even be compared to the adapter and fine-tuning methods, exhibiting negative improvement.

Table 8: **Results of Graph2Graph Transfer in the consideration of GBST and GAST.** : Test ROC-AUC (%) performances on molecular prediction benchmarks with different workflows.

| Pre-train Methods | Tuning Methods | BACE | BBBP | ClinTox | HIV | SIDER | Tox21 | MUV | ToxCast | Avg. | Imp. |
|---|---|---|---|---|---|---|---|---|---|---|---|
| GraphCL | FT | $74.6_{\pm2.2}$ | $68.6_{\pm2.3}$ | $69.8_{\pm2.2}$ | $78.5_{\pm1.2}$ | $59.6_{\pm0.7}$ | $74.4_{\pm0.5}$ | $73.7_{\pm2.7}$ | $62.9_{\pm0.4}$ | 70.3 | – |
| | MetaGP | $72.5_{\pm1.1}$ | $66.9_{\pm1.4}$ | $67.7_{\pm2.5}$ | $77.3_{\pm2.2}$ | $59.0_{\pm1.8}$ | $72.5_{\pm1.4}$ | $74.4_{\pm3.0}$ | $62.2_{\pm0.4}$ | 69.1 | -1.2% |
| | MetaFP | $75.3_{\pm3.6}$ | $66.4_{\pm2.1}$ | $70.3_{\pm1.2}$ | $75.6_{\pm1.3}$ | $59.2_{\pm3.3}$ | $74.4_{\pm0.2}$ | $74.8_{\pm2.8}$ | $63.0_{\pm2.3}$ | 69.9 | -0.4% |
| | Adapter | $76.1_{\pm2.2}$ | $67.8_{\pm1.4}$ | $72.0_{\pm3.8}$ | $77.8_{\pm1.3}$ | $59.6_{\pm1.3}$ | $74.9_{\pm0.9}$ | $75.0_{\pm2.1}$ | $63.1_{\pm0.4}$ | 70.7 | 0.4% |
| | GBST | $73.2_{\pm0.7}$ | $65.1_{\pm0.4}$ | $64.7_{\pm0.1}$ | $70.0_{\pm0.5}$ | $60.6_{\pm0.2}$ | $70.7_{\pm0.1}$ | $74.8_{\pm0.3}$ | $61.4_{\pm0.1}$ | 67.6 | -2.7% |
| | GSST | $79.3_{\pm0.2}$ | $69.5_{\pm1.0}$ | $71.1_{\pm0.4}$ | $72.8_{\pm0.9}$ | $60.6_{\pm0.1}$ | $72.1_{\pm0.1}$ | $78.0_{\pm0.7}$ | $62.9_{\pm0.1}$ | **70.9** | 0.6% |
| SimGRACE | FT | $74.7_{\pm1.0}$ | $65.5_{\pm1.0}$ | $53.8_{\pm2.3}$ | $74.6_{\pm1.2}$ | $58.1_{\pm0.6}$ | $71.9_{\pm0.4}$ | $71.0_{\pm1.9}$ | $61.3_{\pm0.4}$ | 66.3 | – |
| | MetaGP | $72.2_{\pm3.1}$ | $59.8_{\pm1.8}$ | $49.6_{\pm2.5}$ | $69.6_{\pm1.3}$ | $57.7_{\pm2.0}$ | $70.7_{\pm1.7}$ | $71.2_{\pm2.1}$ | $61.6_{\pm2.4}$ | 64.3 | -2.0% |
| | MetaFP | $74.0_{\pm2.3}$ | $62.2_{\pm2.1}$ | $52.3_{\pm3.0}$ | $70.3_{\pm2.6}$ | $58.2_{\pm3.5}$ | $71.9_{\pm1.8}$ | $72.8_{\pm2.7}$ | $61.1_{\pm1.9}$ | 65.4 | -0.9% |
| | Adapter | $74.9_{\pm1.7}$ | $64.6_{\pm1.3}$ | $53.9_{\pm2.0}$ | $72.3_{\pm1.2}$ | $57.2_{\pm0.9}$ | $71.4_{\pm0.6}$ | $71.8_{\pm1.4}$ | $61.3_{\pm0.6}$ | 65.9 | -0.4% |
| | GBST | $65.6_{\pm2.1}$ | $64.1_{\pm1.3}$ | $53.1_{\pm0.7}$ | $69.2_{\pm0.4}$ | $57.7_{\pm0.5}$ | $70.8_{\pm0.1}$ | $72.1_{\pm2.4}$ | $60.6_{\pm0.2}$ | 64.3 | -2.0% |
| | GSST | $73.0_{\pm0.6}$ | $65.4_{\pm0.2}$ | $57.2_{\pm0.3}$ | $69.1_{\pm0.1}$ | $57.9_{\pm0.2}$ | $72.3_{\pm0.3}$ | $74.4_{\pm0.5}$ | $61.6_{\pm0.1}$ | **66.4** | 0.1% |

Nevertheless, GBST and GAST retain their competitiveness in the point cloud transfer task according to Table 11. This observation underscores the distinction between the point cloud classification task, belonging to the category of graph-like data, and other graph tasks. Even when the network struggles to efficiently learn an effective aggregation paradigm in a general way, its impact on the final result is minimal. As a side note, this highlights the ongoing potential for exploration in the realm of transfer learning from the graph domains to graph-like domains.

Table 9: **Results of Node2Node Transfer in the consideration of GBST and GAST.** Test Acc. (%) on diverse node-classification benchmarks with different tuning methods under node-level data pre-training with **error bars**.

| Pre-train Methods | Tuning Methods | Citeseer | | | PubMed | | | Cora | | | Amazon | | | Flickr | | |
|---|---|---|---|---|---|---|---|---|---|---|---|---|---|---|---|---|
| | | GCN | GAT | GIN | GCN | GAT | GIN | GCN | GAT | GIN | GCN | GAT | GIN | GCN | GAT | GIN |
| — | Scratch Train | 64.3±1.1 | 69.2±1.3 | 55.1±0.9 | 75.7±1.1 | 75.1±1.4 | 65.8±0.8 | 76.9±1.0 | 77.0±1.0 | 72.1±0.7 | 92.4±1.3 | 92.3±1.2 | 91.9±0.8 | 53.1±1.3 | 53.0±1.3 | 53.2±0.9 |
| GraphCL | FT | 56.6±1.6 | 56.8±2.0 | 52.8±1.7 | 69.9±1.4 | 70.2±1.5 | 67.3±1.2 | **74.4±1.3** | **73.3±1.8** | 62.4±1.1 | **92.2±2.3** | **92.0±2.5** | **91.0±1.9** | **53.3±1.8** | **52.9±2.2** | **53.9±1.9** |
| | MetaFP | 53.5±2.7 | 55.2±2.5 | 54.5±2.0 | 65.4±1.5 | 68.1±2.4 | 65.2±1.8 | 65.4±1.9 | 67.1±2.0 | 60.8±1.4 | 86.7±2.0 | 87.3±.7 | 82.4±1.1 | 45.5±1.3 | 45.5±1.6 | 44.4±1.3 |
| | Adapter | - | - | 55.2±2.0 | - | - | 65.4±1.7 | - | - | 62.4±1.9 | - | - | 85.3±2.6 | - | - | 50.2±1.6 |
| | GBST | 56.1±2.2 | 53.5±2.7 | 54.4±2.0 | 70.9±2.3 | 68.8±1.9 | 68.6±2.0 | 60.4±2.2 | 59.6±2.2 | 57.1±1.7 | 88.8±2.1 | 87.8±2.3 | 85.3±1.4 | 49.7±2.5 | 45.7±2.6 | 49.7±2.2 |
| | GSST | 54.0±2.0 | 55.8±2.0 | 56.4±1.5 | 69.8±1.6 | 71.8±2.3 | 69.1±1.8 | 63.3±2.1 | 64.0±2.2 | 59.1±2.0 | 88.9±3.0 | 84.8±2.7 | 85.1±2.2 | 49.7±2.3 | 44.3±2.5 | 49.5±2.5 |
| | GAST | 59.7±2.6 | 61.3±2.7 | 54.4±2.1 | 71.1±2.4 | 71.1±2.2 | 69.9±2.4 | 69.6±1.9 | 70.3±2.4 | 57.3±1.7 | 89.2±2.4 | 88.8±2.8 | 86.6±2.4 | 49.7±2.2 | 47.2±2.7 | 49.5±2.0 |
| | GMST | **59.3±1.5** | **63.4±1.9** | **58.8±1.2** | **72.1±2.0** | **75.0±1.7** | **72.6±1.4** | 73.1±2.0 | 72.3±2.2 | **65.4±2.1** | 89.4±1.7 | 90.2±1.8 | 86.2±2.1 | 51.9±2.2 | 47.7±2.4 | 49.9±2.1 |
| SimGRACE | FT | 58.9±1.2 | 57.6±1.1 | 45.5±0.7 | 71.3±1.9 | 71.7±2.2 | 64.1±1.3 | 72.9±1.4 | 71.2±2.0 | 64.4±1.0 | **92.4±2.3** | **92.3±2.7** | **91.3±2.3** | **53.6±2.5** | **50.8±2.5** | **53.8±2.1** |
| | MetaFP | 54.2±3.0 | 55.3±3.3 | 46.6±2.9 | 67.2±2.5 | 68.5±2.7 | 65.7±2.1 | 66.3±2.5 | 63.4±2.8 | 60.2±2.4 | 83.5±2.3 | 85.4±2.8 | 80.5±1.7 | 47.7±2.4 | 43.6±2.4 | 48.8±2.3 |
| | Adapter | - | - | 48.4±1.7 | - | - | 63.2±2.4 | - | - | 61.8±2.2 | - | - | 80.2±1.9 | - | - | 51.2±2.7 |
| | GBST | 53.0±2.3 | 48.8±2.5 | 47.2±1.9 | 69.7±2.3 | 69.4±2.7 | 64.8±2.2 | 62.3±1.9 | 59.6±2.5 | 51.9±1.5 | 88.9±2.2 | 88.6±1.9 | 85.7±1.5 | 49.1±2.3 | 45.8±2.5 | 49.7±1.8 |
| | GSST | 52.0±1.9 | 52.1±2.2 | 49.5±1.8 | 68.0±2.7 | 70.0±2.5 | 67.3±1.9 | 64.6±1.9 | 59.3±2.7 | 53.9±2.3 | 88.8±2.4 | 87.9±2.7 | 80.3±1.5 | 48.9±2.3 | 45.2±2.5 | 49.7±1.8 |
| | GAST | 54.3±2.4 | 51.3±2.7 | 47.8±2.0 | 69.9±1.9 | 71.8±2.4 | 63.8±2.0 | 63.6±2.6 | 63.6±3.0 | 51.2±2.2 | 88.9±3.0 | 89.4±3.3 | 85.5±2.7 | 49.2±2.3 | 46.7±3.5 | 49.7±3.0 |
| | GMST | **61.6±2.1** | **63.4±2.4** | **58.9±2.0** | **73.2±1.7** | **75.8±2.3** | **72.7±1.5** | **75.1±2.1** | **72.2±2.4** | **66.7±1.7** | 90.9±2.5 | 90.5±3.0 | 84.2±2.1 | 50.6±2.8 | 47.7±3.2 | 51.2±2.7 |

Table 10: **Results of Graph2Node Transfer in the consideration of GBST and GAST.** Test Acc. (%) on diverse node-classification benchmarks with different tuning methods under graph-level data pre-training with **error bars.**

| Pre-train Methods | Tuning Methods | Citeseer | | | PubMed | | | Cora | | | Amazon | | | Flickr | | |
|---|---|---|---|---|---|---|---|---|---|---|---|---|---|---|---|---|
| | | GCN | GAT | GIN | GCN | GAT | GIN | GCN | GAT | GIN | GCN | GAT | GIN | GCN | GAT | GIN |
| — | Scratch Train | 64.3±1.1 | 69.2±1.3 | 55.1±0.9 | 75.7±1.1 | 75.1±1.4 | 65.8±0.8 | 76.9±1.0 | 77.0±1.0 | 72.1±0.7 | 92.4±1.3 | 92.3±1.2 | 91.9±0.8 | 53.1±1.3 | 53.0±1.3 | 53.2±0.9 |
| GraphCL | FT | 52.6±2.6 | 49.9±2.8 | 46.9±2.1 | 68.6±2.0 | 68.0±2.1 | 63.5±2.3 | 69.2±2.5 | 60.9±2.5 | 63.1±1.8 | **91.8±2.4** | 89.1±2.7 | **90.4±1.8** | **52.3±2.7** | 49.9±2.7 | **52.4±2.5** |
| | MetaFP | 50.4±2.0 | 49.8±2.3 | 45.5±2.3 | 65.9±2.4 | 66.3±2.8 | 60.4±2.1 | 64.3±2.4 | 61.0±2.6 | 60.3±2.0 | 85.5±2.8 | 86.1±3.1 | 80.4±2.5 | 48.4±2.5 | 44.8±3.1 | 42.6±2.5 |
| | Adapter | - | - | 46.1±2.2 | - | - | 59.4±2.5 | - | - | 57.8±2.4 | - | - | 82.8±2.3 | - | - | 48.7±2.0 |
| | GBST | 49.0±2.1 | 48.4±2.4 | 50.9±1.7 | 64.4±2.0 | 62.4±1.8 | 67.1±2.1 | 51.6±2.7 | 50.0±2.8 | 55.0±2.5 | 87.0±2.7 | 85.6±2.3 | 84.8±2.1 | 48.2±2.5 | 45.7±3.0 | 48.0±2.5 |
| | GSST | 48.7±2.4 | 49.9±2.6 | 50.2±2.4 | 64.6±2.3 | 64.4±2.3 | 65.8±1.9 | 52.2±2.6 | 51.3±2.8 | 56.0±2.4 | 83.7±2.5 | 80.8±3.2 | 80.2±2.1 | 48.2±2.5 | 44.5±2.4 | 47.6±2.5 |
| | GAST | 48.4±2.2 | 50.5±2.5 | 50.1±2.0 | 68.8±2.4 | 64.9±2.4 | 68.8±2.3 | 58.9±2.5 | 52.1±3.0 | 55.5±2.6 | 88.4±2.9 | 87.5±3.0 | 84.7±2.7 | 50.3±2.4 | 46.1±2.2 | 47.4±2.4 |
| | GMST | **61.9±2.7** | **62.4±2.7** | **57.9±2.4** | **73.1±2.5** | **73.7±2.8** | **73.9±2.1** | **74.8±1.9** | **72.3±2.6** | **66.5±1.5** | 88.6±2.4 | **89.8±2.8** | 85.5±2.5 | 51.3±2.4 | 47.3±2.4 | 49.5±2.1 |
| SimGRACE | FT | 53.7±1.8 | 53.0±2.5 | 43.3±1.7 | 59.3±2.0 | 68.1±2.2 | 61.8±2.1 | 65.0±2.3 | 64.3±2.5 | 57.7±2.0 | **92.3±2.8** | 89.5±3.3 | **90.9±2.5** | **52.4±2.5** | 48.1±2.8 | **53.5±2.3** |
| | MetaFP | 51.3±2.6 | 52.3±2.8 | 44.1±1.9 | 56.4±2.4 | 61.4±2.2 | 56.3±2.0 | 61.1±1.8 | 64.5±2.1 | 56.8±2.2 | 85.2±2.7 | 86.5±2.9 | 81.8±2.1 | 46.6±2.6 | 47.0±2.6 | 44.4±2.0 |
| | Adapter | - | - | 45.4±2.3 | - | - | 59.9±2.4 | - | - | 56.6±2.1 | - | - | 80.4±2.5 | - | - | 46.3±2.7 |
| | GBST | 48.5±2.3 | 46.0±2.5 | 50.2±2.3 | 60.7±2.7 | 61.3±3.0 | 66.4±2.3 | 53.3±2.7 | 49.9±2.7 | 49.6±1.9 | 85.1±2.5 | 86.2±2.3 | 83.9±2.1 | 48.0±2.8 | 46.0±2.8 | 47.2±2.4 |
| | GSST | 49.9±2.4 | 45.4±2.6 | 51.9±1.7 | 58.2±2.4 | 61.6±2.9 | 65.3±2.1 | 54.5±2.9 | 49.2±2.8 | 48.3±2.4 | 81.0±2.5 | 81.7±2.7 | 77.4±2.4 | 47.8±2.5 | 44.9±2.3 | 48.9±2.3 |
| | GAST | 48.2±2.4 | 45.1±3.0 | 48.0±2.2 | 61.6±2.8 | 65.8±2.8 | 66.2±2.1 | 56.3±2.2 | 52.7±3.2 | 48.7±2.0 | 89.3±3.1 | 89.1±3.4 | 85.4±2.5 | 48.7±2.3 | 46.7±3.1 | 49.0±2.4 |
| | GMST | **63.0±2.8** | **62.2±3.3** | **58.5±2.8** | **72.7±2.3** | **74.8±2.4** | **73.3±1.8** | **74.3±2.4** | **71.1±2.8** | **65.2±2.5** | 89.7±2.2 | **89.5±3.1** | 85.3±2.0 | 50.2±2.2 | 47.3±2.8 | 51.6±2.1 |

Table 11: **Results of Graph2PtCld Transfer.** We configure pre-trained models on graph-level and node-level data across distinct types of graph layers as the initialization for backbone.

| Pre-train Methods | Tuning Methods | Graph-level: ogbg-molhiv | | | Node-level: ogbn-arxiv | | |
|---|---|---|---|---|---|---|---|
| Backbones | | GCN | GAT | GIN | GCN | GAT | GIN |
| — | Scratch Train | 77.4±1.8 | 81.6±1.1 | 87.9±1.0 | 77.4±2.1 | 81.6±1.5 | 87.9±1.3 |
| GraphCL | FT | 78.1±1.7 | 74.9±2.1 | 83.9±1.1 | 73.4±3.0 | **75.4±2.4** | **87.3±3.1** |
| | MetaFP | 73.1±3.6 | 70.3±2.4 | 78.8±1.6 | 70.2±2.5 | 71.9±2.8 | 74.1±2.0 |
| | Adapter | - | - | 80.9±2.2 | - | - | 75.9±2.8 |
| | GBST | 76.6±1.9 | **79.2±2.6** | 83.5±1.8 | 73.8±3.4 | 74.6±3.7 | 86.0±2.4 |
| | GAST | 79.3±1.6 | 78.2±2.0 | 82.9±2.4 | **76.4±3.5** | 72.1±2.9 | 83.8±2.5 |
| | GSST | **81.7±2.9** | 78.6±3.7 | **84.7±2.8** | 70.4±3.1 | 74.0±2.7 | 80.4±3.8 |
| | GMST | 77.8±2.6 | 74.8±3.5 | 82.5±2.4 | 75.6±3.4 | 74.5±2.7 | 80.8±2.7 |
| SimGRACE | FT | 78.6±1.5 | 70.9±1.8 | 77.3±1.7 | 72.8±2.0 | 78.0±2.3 | **79.7±1.9** |
| | MetaFP | 71.7±1.2 | 66.9±2.2 | 76.5±1.5 | 69.8±2.3 | 71.0±1.7 | 74.4±2.0 |
| | Adapter | - | - | 76.9±3.2 | - | - | 73.8±2.9 |
| | GBST | 75.7±2.8 | 77.4±2.5 | 75.4±1.7 | 72.1±3.9 | 76.8±3.0 | 79.3±2.7 |
| | GAST | 76.9±2.4 | 74.3±2.6 | 74.1±1.5 | 73.3±2.8 | 77.5±3.1 | 79.3±2.6 |
| | GSST | **79.3±4.2** | **81.8±3.7** | **77.3±2.3** | 71.8±3.8 | 69.2±2.8 | 63.9±2.0 |
| | GMST | 65.2±2.7 | 65.1±3.1 | 76.4±2.7 | **78.1±3.6** | **78.1±3.0** | 78.7±3.2 |

## A.3 ADDITIONAL ABLATION STUDIES

To validate the robustness of our method, we conducted additional ablation experiments in two key areas:

• **Influence of Source Dataset.** Confirming the algorithm's ability to achieve comparable performance on pre-trained models trained on different datasets.

- **Influence of GNN Architecture.** Verifying that the algorithm can maintain stable prediction performance across backbone architectures with varying numbers of layers.

- **Influence of Pre-training Methods.** Proving that our framework can flexibly adopt different graph-level pre-training methods and maintain stable prediction performance during tuning stage.

- **Influence of Side Network Structure.** Justifying the use of MLP as a side network for both the GSST GMST tuning algorithm.

Table 12: **Ablation Study: Node2Node Transfer Results for Model Pre-trained on Flickr.** Test Acc. (%) on diverse node-classification benchmarks with different tuning methods under node-level data pre-training.

| Pre-train Methods | Tuning Methods | Citeseer | | | PubMed | | | Cora | | | Amazon | | | ogbn-arxiv | | |
|---|---|---|---|---|---|---|---|---|---|---|---|---|---|---|---|---|
| | | GCN | GAT | GIN | GCN | GAT | GIN | GCN | GAT | GIN | GCN | GAT | GIN | GCN | GAT | GIN |
| —- | Scratch Train | 64.30 | 69.21 | 55.10 | 75.70 | 75.10 | 65.80 | 76.90 | 77.00 | 72.10 | 92.37 | 92.33 | 91.89 | 61.36 | 63.63 | 64.37 |
| GraphCL | FT | 61.20 | 50.90 | 52.30 | 72.00 | 61.60 | 60.50 | 72.20 | 64.10 | 65.50 | 91.78 | 88.00 | 90.84 | 59.65 | 61.08 | 58.78 |
| | GBST | 53.30 | 45.80 | 46.70 | 67.00 | 64.60 | 65.50 | 57.00 | 54.30 | 52.20 | 88.62 | 85.61 | 86.30 | 35.46 | 31.12 | 30.52 |
| | GSST | 54.40 | 47.50 | 48.10 | 66.30 | 64.10 | 60.00 | 63.40 | 50.90 | 53.70 | 88.19 | 81.86 | 82.99 | 31.16 | 26.94 | 25.06 |
| | GAST | 54.10 | 55.30 | 46.40 | 67.60 | 64.00 | 63.30 | 61.40 | 60.00 | 50.00 | 87.93 | 88.77 | 86.15 | 38.95 | 43.46 | 31.38 |
| | GMST | 59.60 | 62.80 | 56.30 | 72.90 | 75.40 | 72.20 | 74.70 | 70.40 | 64.40 | 89.51 | 88.95 | 89.86 | 35.27 | 35.46 | 41.85 |
| SimGRACE | FT | 57.90 | 52.40 | 50.10 | 70.00 | 67.10 | 62.00 | 71.60 | 65.90 | 65.40 | 91.60 | 90.11 | 91.20 | 60.71 | 61.92 | 60.67 |
| | GBST | 50.30 | 46.50 | 49.2 | 67.40 | 62.90 | 63.90 | 56.90 | 52.00 | 54.10 | 88.40 | 85.42 | 85.50 | 39.67 | 28.69 | 38.75 |
| | GSST | 53.50 | 46.70 | 50.90 | 64.60 | 60.70 | 58.30 | 58.00 | 52.40 | 52.30 | 88.55 | 81.02 | 83.46 | 35.41 | 25.16 | 37.12 |
| | GAST | 53.70 | 53.20 | 50.40 | 68.40 | 65.50 | 66.20 | 64.00 | 62.40 | 54.00 | 88.00 | 88.51 | 85.53 | 40.55 | 45.79 | 30.64 |
| | GMST | 60.20 | 63.50 | 57.20 | 70.30 | 75.60 | 71.30 | 73.70 | 70.30 | 64.50 | 89.99 | 89.88 | 86.59 | 34.97 | 44.35 | 34.35 |

Table 13: **Ablation Study: Graph2Node Transfer Results for Model Pre-trained on MUV.** Test Acc. (%) on diverse node-classification benchmarks with different tuning methods under graph-level data pre-training.

| Pre-train Methods | Tuning Methods | Citeseer | | | PubMed | | | Cora | | | Amazon | | | Flickr | | |
|---|---|---|---|---|---|---|---|---|---|---|---|---|---|---|---|---|
| | | GCN | GAT | GIN | GCN | GAT | GIN | GCN | GAT | GIN | GCN | GAT | GIN | GCN | GAT | GIN |
| — | Scratch Train | 64.30 | 69.21 | 55.10 | 75.70 | 75.10 | 65.80 | 76.90 | 77.00 | 72.10 | 92.37 | 92.33 | 91.89 | 53.07 | 52.97 | 53.15 |
| GraphCL | FT | 54.60 | 46.00 | 53.90 | 65.10 | 68.70 | 68.70 | 69.50 | 72.10 | 68.40 | 91.78 | 91.31 | 90.91 | 52.99 | 49.35 | 53.23 |
| | GBST | 51.00 | 57.70 | 50.30 | 63.10 | 64.60 | 66.30 | 52.20 | 49.90 | 53.10 | 87.35 | 86.11 | 84.26 | 46.28 | 45.78 | 48.72 |
| | GSST | 49.00 | 46.50 | 50.20 | 57.20 | 63.70 | 66.80 | 53.20 | 49.80 | 54.70 | 84.30 | 82.19 | 79.93 | 45.50 | 44.94 | 48.44 |
| | GAST | 48.10 | 50.90 | 50.10 | 66.30 | 66.40 | 68.00 | 56.10 | 60.20 | 54.20 | 87.93 | 88.80 | 83.93 | 50.10 | 46.02 | 49.63 |
| | GMST | 59.10 | 60.60 | 55.20 | 71.30 | 72.50 | 71.80 | 74.20 | 72.50 | 69.90 | 90.48 | 90.55 | 88.01 | 51.41 | 48.60 | 51.42 |
| SimGRACE | FT | 52.40 | 52.60 | 48.00 | 69.30 | 66.90 | 61.30 | 69.10 | 65.80 | 63.70 | 91.60 | 90.19 | 90.40 | 53.06 | 49.62 | 53.26 |
| | GBST | 45.70 | 46.50 | 45.40 | 67.00 | 66.70 | 61.80 | 49.50 | 48.40 | 46.00 | 85.71 | 85.50 | 83.90 | 46.19 | 45.54 | 45.84 |
| | GSST | 46.60 | 43.80 | 44.10 | 64.00 | 63.80 | 62.80 | 53.10 | 50.90 | 45.80 | 81.90 | 80.48 | 75.50 | 45.54 | 44.09 | 45.54 |
| | GAST | 44.60 | 50.80 | 47.90 | 68.60 | 68.00 | 60.80 | 52.70 | 59.00 | 48.80 | 88.11 | 89.24 | 84.62 | 49.70 | 47.36 | 47.04 |
| | GMST | 58.20 | 56.40 | 57.90 | 72.70 | 74.00 | 70.60 | 74.00 | 72.70 | 66.10 | 90.44 | 90.02 | 85.90 | 51.42 | 46.15 | 52.80 |

### A.3.1 THE EFFECT OF DIFFERENT PRE-TRAINING DATA ON THE ARBITRARY GRAPH TRANSFER

To assess the impact of different pre-training datasets on the final transfer performance of our proposed method, we conducted new experiments in two task scenarios: medium and hard. For Node2Node Transfer, we chose Flickr as the pre-training dataset and utilized all other graph classification datasets as downstream tasks to evaluate transfer learning performance. Conversely, for Graph2Node transfer, we opted for the MUV dataset as the training data and selected the same graph classification datasets as those applied in the main text as downstream tasks for transfer learning. The rest of the experimental setup is consistent with the main text.

The results of the experiments are presented in Table 12, where our method performs well on the first 4 test datasets in a moderately difficult task but exhibits poor performance on the ogbn-arxiv transfer. It is noteworthy that both the number of nodes and the number of edges in the Flickr dataset are only about half of those in ogbn-arxiv. This suggests that the transfer of downstream tasks can face challenges when the knowledge from the source domain is not sufficiently rich because the absence of knowledge in the source domain can lead to pre-trained models being unable to achieve sufficiently generalized performance. This phenomenon aligns with the pre-train-tuning

paradigm. Examining the remaining results in the table, it is evident that our method maintains stable performance. The algorithm effectively carries out downstream task transfer learning on a well-pre-trained model.

### A.3.2 THE EFFECT OF DIFFERENT BACKBONE LAYERS ON THE ARBITRARY GRAPH TRANSFER

To investigate the impact of different numbers of backbone layers on the performance of the GSST and GMST methods, we configured the number of layers of the backbone to 5, the maximum currently used in stacked graph neural networks. Subsequently, we trained and tested these configurations on medium and difficult task scenarios, utilizing the same pre-trained models and downstream tasks as detailed in the main text. The results obtained are recorded in Tables 14, 15.

Table 14: **Ablation Study: Node2Node Transfer Results with 5-layers Backbone.** Test Acc. (%) on diverse node-classification benchmarks with different tuning methods under node-level data pre-training.

| Pre-train Methods | Tuning Methods | Citeseer | | | PubMed | | | Cora | | | Amazon | | | Flickr | | |
|---|---|---|---|---|---|---|---|---|---|---|---|---|---|---|---|---|
| | | GCN | GAT | GIN | GCN | GAT | GIN | GCN | GAT | GIN | GCN | GAT | GIN | GCN | GAT | GIN |
| — | Scratch Train | 63.20 | 65.80 | 58.20 | 73.50 | 74.50 | 70.90 | 76.00 | 77.40 | 69.80 | 91.97 | 91.75 | 91.13 | 55.76 | 54.02 | 54.89 |
| GraphCL | FT | 62.10 | 65.30 | 48.70 | 70.30 | 70.80 | 66.00 | 75.40 | 74.80 | 68.80 | 91.56 | 90.62 | 90.19 | 55.23 | 55.77 | 53.56 |
| | GBST | 52.30 | 58.70 | 47.30 | 68.10 | 69.30 | 66.40 | 64.30 | 61.20 | 50.90 | 87.97 | 87.13 | 84.41 | 48.52 | 48.53 | 48.67 |
| | GSST | 49.30 | 55.00 | 46.70 | 70.50 | 71.80 | 68.10 | 66.00 | 67.30 | 51.60 | 87.42 | 84.59 | 73.79 | 49.72 | 48.33 | 48.83 |
| | GAST | 54.00 | 54.40 | 50.20 | 70.10 | 69.70 | 68.80 | 64.80 | 66.20 | 52.90 | 88.08 | 87.53 | 84.73 | 48.67 | 49.75 | 48.80 |
| | GMST | 62.50 | 64.40 | 50.50 | 73.80 | 72.90 | 71.60 | 71.00 | 72.00 | 69.10 | 88.46 | 89.90 | 86.41 | 51.44 | 49.96 | 50.97 |
| SimGRACE | FT | 57.50 | 56.20 | 47.30 | 71.40 | 73.80 | 66.30 | 74.00 | 70.60 | 58.30 | 91.38 | 91.42 | 89.68 | 54.87 | 55.22 | 54.46 |
| | GBST | 52.90 | 52.80 | 50.50 | 66.50 | 67.30 | 61.90 | 52.80 | 60.50 | 49.60 | 87.64 | 88.55 | 83.86 | 48.21 | 47.11 | 48.65 |
| | GSST | 45.30 | 47.70 | 40.20 | 65.90 | 65.70 | 65.60 | 53.90 | 58.50 | 36.20 | 87.46 | 86.80 | 80.74 | 48.41 | 47.87 | 49.73 |
| | GAST | 55.80 | 56.40 | 49.80 | 68.60 | 67.50 | 65.30 | 58.20 | 59.20 | 49.50 | 88.08 | 88.33 | 85.06 | 48.76 | 48.61 | 49.55 |
| | GMST | 56.40 | 57.2 | 50.30 | 70.50 | 73.80 | 66.90 | 69.20 | 69.80 | 53.60 | 88.74 | 86.01 | 87.90 | 50.88 | 49.75 | 50.78 |

Table 15: **Ablation Study: Graph2Node Transfer Results with 5-layers Backbone.** Test Acc. (%) on diverse node-classification benchmarks with different tuning methods under graph-level data pre-training.

| Pre-train Methods | Tuning Methods | Citeseer | | | PubMed | | | Cora | | | Amazon | | | Flickr | | |
|---|---|---|---|---|---|---|---|---|---|---|---|---|---|---|---|---|
| | | GCN | GAT | GIN | GCN | GAT | GIN | GCN | GAT | GIN | GCN | GAT | GIN | GCN | GAT | GIN |
| — | Scratch Train | 63.20 | 65.80 | 58.20 | 74.80 | 76.00 | 68.70 | 76.00 | 77.40 | 69.80 | 91.97 | 91.75 | 91.13 | 55.76 | 54.02 | 54.89 |
| GraphCL | FT | 53.10 | 51.70 | 50.80 | 68.10 | 68.80 | 68.30 | 70.40 | 70.70 | 63.50 | 90.28 | 90.22 | 89.53 | 51.52 | 52.88 | 50.29 |
| | GBST | 51.30 | 51.10 | 47.20 | 69.60 | 65.60 | 63.50 | 52.20 | 55.20 | 56.70 | 85.35 | 84.88 | 84.95 | 48.23 | 47.44 | 48.99 |
| | GSST | 41.80 | 42.70 | 39.70 | 63.20 | 57.20 | 66.30 | 46.20 | 54.10 | 48.00 | 81.86 | 75.91 | 82.92 | 49.78 | 47.96 | 49.83 |
| | GAST | 51.00 | 54.00 | 50.00 | 69.80 | 70.40 | 66.50 | 55.90 | 64.20 | 57.80 | 86.99 | 85.37 | 84.70 | 49.98 | 49.28 | 49.80 |
| | GMST | 58.60 | 58.40 | 56.80 | 71.20 | 74.70 | 69.60 | 70.90 | 72.40 | 60.90 | 89.70 | 89.78 | 88.92 | 51.19 | 49.29 | 50.25 |
| SimGRACE | FT | 54.40 | 55.10 | 49.50 | 66.00 | 69.30 | 65.80 | 68.90 | 66.80 | 62.30 | 90.60 | 90.42 | 90.37 | 50.67 | 51.45 | 51.22 |
| | GBST | 46.06 | 48.10 | 51.00 | 63.90 | 64.10 | 63.40 | 51.10 | 47.90 | 51.70 | 85.64 | 85.97 | 85.21 | 47.43 | 46.28 | 48.56 |
| | GSST | 32.90 | 39.80 | 41.20 | 63.10 | 62.70 | 56.80 | 43.40 | 37.60 | 38.80 | 81.42 | 72.66 | 80.56 | 49.76 | 47.21 | 48.53 |
| | GAST | 48.60 | 49.50 | 50.20 | 63.30 | 67.30 | 64.70 | 51.40 | 48.50 | 48.20 | 86.30 | 84.44 | 84.44 | 47.98 | 48.33 | 49.80 |
| | GMST | 55.50 | 55.40 | 50.50 | 69.00 | 74.20 | 67.00 | 70.70 | 70.40 | 66.40 | 89.53 | 88.62 | 88.28 | 50.49 | 48.89 | 50.00 |

The experimental results in Tables 14, 15 demonstrate that even with the backbone layers set to 5, our GMST algorithm consistently achieves impressive performance across various task scenarios, pre-training methods, and graph convolution layers. This underscores the stability of our algorithm.

In summary, our method exhibits robustness across various transfer scenarios and experimental setups.

### A.3.3 ADAPTATION OF THE PRE-TRAINING STAGE TO DIFFERENT GRAPH-LEVEL PRE-TRAINING METHODS

As described in the main paper, our GraphBridge framework is capable of adapting various graph-level pre-training methods to pre-train our base model in Pre-training Stage. Consequently, we also evaluated the the transfer performance following pre-training with the GCC(Qiu et al., 2020) and GPT-GNN(Hu et al., 2020b) methods.

The GCC(Qiu et al., 2020) framework is a graph-level pre-training methods which learns structural representations across graphs by leveraging the idea of contrastive learning to design the graph pre-training task as instance discrimination. Its basic idea is to sample instances from input graphs, treat each of them as a distinct class of its own, and learn to encode and discriminate between these instances. Similar to GraphCL and SimGRACE, GCC employs graph-level GNN pre-training schemes based on contrastive learning strategies. In our experiments, we do not utilize the end-to-end fine-tuning setup proposed in GCC's paper. Instead, we focus solely on the pre-training component of GCC to obtain our base models for our arbitrary transfer learning.

On the other hand, GPT-GNN(Hu et al., 2020b) is a generative pre-training framework for graph neural networks based on the self-supervised attributed graph generation task proposed by the author, with which both the structure and attributes of the graph are modeled. In attributed graph generation task, the graph generation objective has been decomposed into two components: Attribute Generation and Edge Generation, whose joint optimization is equivalent to maximizing the probability likelihood of the whole attributed graph. In doing this, the pre-trained model can capture the general knowledge of the graphs. Here, we only utilize the GPT-GNN pre-training stage as well to obtain our base models.

Table 16: **Ablation Study: Node2Node Transfer Results with the Base Model Pre-trained by Different Graph-level Pre-training Methods.** Test Acc. (%) on diverse node-classification benchmarks with different tuning methods under graph-level data pre-training.

| Pre-train Methods | Tuning Methods | Citeseer | | | PubMed | | | Cora | | | Amazon | | | Flickr | | |
|---|---|---|---|---|---|---|---|---|---|---|---|---|---|---|---|---|
| | | GCN | GAT | GIN | GCN | GAT | GIN | GCN | GAT | GIN | GCN | GAT | GIN | GCN | GAT | GIN |
| — | Scratch Train | 64.30 | 69.20 | 55.10 | 75.70 | 75.10 | 65.80 | 76.90 | 77.00 | 72.10 | 92.37 | 92.33 | 91.89 | 53.07 | 52.97 | 53.15 |
| GCC | FT | 58.10 | 56.40 | 46.80 | 70.80 | 70.70 | 65.20 | 72.90 | 71.40 | 61.60 | **91.44** | **92.68** | **90.87** | **52.79** | **50.61** | **53.24** |
| | GSST | 50.40 | 51.00 | 50.10 | 66.70 | 69.30 | 67.80 | 63.20 | 60.10 | 54.70 | 87.56 | 87.23 | 81.77 | 48.15 | 43.56 | 49.21 |
| | GMST | **61.90** | **62.80** | **58.60** | **73.10** | **75.40** | **71.50** | **73.50** | **71.20** | **67.29** | 90.37 | 90.10 | 83.87 | 50.22 | 45.62 | 51.46 |
| GPT-GNN | FT | 54.20 | 51.30 | 42.40 | 67.60 | 65.20 | 62.40 | 70.90 | 69.40 | 62.10 | **91.42** | **92.12** | **90.22** | **52.77** | **50.41** | **52.89** |
| | GSST | 48.70 | 50.10 | 43.10 | 64.40 | 66.70 | 63.70 | 61.70 | 57.40 | 51.80 | 85.78 | 83.23 | 77.42 | 44.98 | 45.12 | 47.42 |
| | GMST | **56.10** | **56.20** | **54.70** | **70.10** | **70.80** | **68.50** | **74.70** | **71.00** | **63.20** | 87.84 | 88.13 | 82.56 | 47.33 | 47.18 | 50.03 |

Table 17: **Ablation Study: Graph2Node Transfer Results with the Base Model Pre-trained by Different Graph-level Pre-training Methods.** Test Acc. (%) on diverse node-classification benchmarks with different tuning methods under graph-level data pre-training.

| Pre-train Methods | Tuning Methods | Citeseer | | | PubMed | | | Cora | | | Amazon | | | Flickr | | |
|---|---|---|---|---|---|---|---|---|---|---|---|---|---|---|---|---|
| | | GCN | GAT | GIN | GCN | GAT | GIN | GCN | GAT | GIN | GCN | GAT | GIN | GCN | GAT | GIN |
| — | Scratch Train | 64.20 | 69.20 | 55.10 | 75.70 | 75.10 | 65.80 | 76.90 | 77.00 | 72.10 | 92.37 | 92.33 | 91.89 | 53.07 | 52.97 | 53.15 |
| GCC | FT | 57.70 | 56.10 | 47.10 | 69.70 | 70.30 | 64.70 | 71.80 | 71.70 | 60.70 | 90.21 | 91.23 | **90.14** | **51.46** | **50.21** | **52.19** |
| | GSST | 51.60 | 52.10 | 50.70 | 65.20 | 69.50 | 67.10 | 61.90 | 60.40 | 54.70 | 86.11 | 85.67 | 80.25 | 46.35 | 42.78 | 47.96 |
| | GMST | **61.40** | **61.90** | **56.60** | **71.30** | **74.20** | **70.50** | **72.60** | **71.00** | **65.20** | **90.35** | **91.24** | 83.76 | 50.53 | 46.82 | 51.32 |
| GPT-GNN | FT | 54.10 | 50.60 | 42.10 | 67.30 | 68.70 | 62.10 | 70.40 | 70.20 | 62.00 | 89.12 | 90.13 | **90.47** | **51.46** | **50.20** | **51.77** |
| | GSST | 47.20 | 50.70 | 44.30 | 64.00 | 66.00 | 64.30 | 63.80 | 57.00 | 51.30 | 86.75 | 85.83 | 80.11 | 44.93 | 44.40 | 46.32 |
| | GMST | **56.20** | **57.80** | **55.90** | **70.20** | **71.80** | **69.70** | **72.10** | **71.20** | **63.70** | **89.88** | **90.63** | 82.66 | 48.22 | 47.21 | 50.07 |

The supplemental experiments were conducted on the medium and difficult task scenarios as well, with all other settings for the parameters consistent with those in the main paper experiment. The results of the experiment are presented in Tables 16 and 17.

According to the results presented in the Tables 16 and 17, our framework still has the ability to perform arbitrary end-to-end graph transfer learning effectively using different graph-level pre-training methods in the Pre-training Stage. Although different pre-training strategies influence the absolute transfer learning performance, our proposed GMST structure consistently achieves superior performance on most downstream tasks and backbones, aligning with the results reported in the main paper's experiments.

In summary, the results of the ablation experiments displayed in this section demonstrate the flexibility and adaptability of the GraphBridge framework during both the pre-training and fine-tuning stages, highlighting its practical value.

### A.3.4 THE EFFECT OF DIFFERENT SIDE NETWORK STRUCTURES ON THE TUNING RESULTS

To investigate the impact of side network structures on arbitrary graph transfer, we conducted additional experiments with GMST fine-tuning. Specifically, we used each GNN backbone's corresponding lightweight structure as the side network (consistent with the implementation of Sung et al. (2022a)) to conduct the GMST across different datasets and compared the results with those obtained using an MLP as the side network. These experiments were carried out under the Middle and Hard task scenarios, and the results are presented in Table 18 and Table 19, respectively.

Table 18: **Ablation Study: Node2Node Transfer Results with Different Side Network Structures using GMST Tuning Algorithm.** Test Acc. (%) on diverse node-classification benchmarks with different tuning methods under graph-level data pre-training.

| Pre-train Methods | Tuning Methods | Citeseer | | | PubMed | | | Cora | | | Amazon | | | Flickr | | |
|---|---|---|---|---|---|---|---|---|---|---|---|---|---|---|---|---|
| | | GCN | GAT | GIN | GCN | GAT | GIN | GCN | GAT | GIN | GCN | GAT | GIN | GCN | GAT | GIN |
| GraphCL | GNN-Side | 60.10 | 63.20 | 60.50 | 71.80 | 74.70 | 72.60 | 73.50 | 71.70 | 65.80 | 89.88 | 89.93 | 86.72 | 51.99 | 47.56 | 50.37 |
| | MLP-Side | 59.30 | 63.40 | 58.80 | 72.10 | 75.00 | 72.60 | 73.10 | 72.30 | 65.40 | 89.42 | 90.19 | 86.15 | 51.92 | 47.70 | 49.94 |
| SimGRACE | GNN-Side | 61.90 | 64.00 | 60.00 | 73.00 | 75.60 | 72.70 | 74.90 | 72.00 | 66.90 | 90.78 | 90.32 | 84.85 | 50.67 | 47.57 | 52.08 |
| | MLP-Side | 61.60 | 63.40 | 58.90 | 73.20 | 75.80 | 72.70 | 75.10 | 72.20 | 66.70 | 90.88 | 90.53 | 84.19 | 50.56 | 47.71 | 51.16 |

Table 19: **Ablation Study: Graph2Node Transfer Results with Different Side Network Structures using GMST Tuning Algorithm.** Test Acc. (%) on diverse node-classification benchmarks with different tuning methods under graph-level data pre-training.

| Pre-train Methods | Tuning Methods | Citeseer | | | PubMed | | | Cora | | | Amazon | | | Flickr | | |
|---|---|---|---|---|---|---|---|---|---|---|---|---|---|---|---|---|
| | | GCN | GAT | GIN | GCN | GAT | GIN | GCN | GAT | GIN | GCN | GAT | GIN | GCN | GAT | GIN |
| GraphCL | GNN-Side | 62.10 | 62.40 | 58.50 | 73.00 | 73.50 | 74.20 | 74.50 | 72.50 | 66.90 | 88.75 | 89.93 | 85.76 | 51.42 | 47.12 | 49.84 |
| | MLP-Side | 61.90 | 62.40 | 57.90 | 73.10 | 73.70 | 73.90 | 74.80 | 72.30 | 66.50 | 88.62 | 89.79 | 85.50 | 51.30 | 47.34 | 49.54 |
| SimGRACE | GNN-Side | 63.30 | 62.50 | 58.80 | 72.40 | 74.80 | 73.90 | 74.60 | 71.00 | 65.90 | 89.54 | 89.68 | 85.76 | 50.57 | 47.12 | 51.92 |
| | MLP-Side | 63.00 | 62.20 | 58.50 | 72.70 | 74.80 | 73.30 | 74.30 | 71.10 | 65.20 | 89.67 | 89.46 | 85.26 | 50.24 | 47.29 | 51.57 |

The experimental results demonstrate that employing the corresponding lightweight GNN as the side network does not yield significant performance improvements for GMST tuning in either the Middle or Hard task scenarios. Furthermore, based on the comparative analysis of training efficiency between GNN and MLP in Sung et al. (2022a) and Han et al. (2022), using an MLP as the side network ensures that computational overhead increases linearly with data scale, thereby maintaining the efficiency of our GMST algorithm. In conclusion, our innovative use of an MLP as the side network for pre-trained GNNs significantly enhances fine-tuning efficiency while preserving performance in arbitrary graph domain transfer, which is a successful attempt.

### A.4 SUPPLEMENTARY TRANSFER SCENARIOS

To further refine our task setup and comprehensively validate the generalization of the framework, we conducted additional transfer experiments on scenarios with the same level of difficulty as the Graph2Node task, including Node2Graph and Graph2Edge scenarios.

• **Node2Graph.** Transfer learning from node classification tasks to graph classification tasks within unrelated knowledge domains.

• **Graph2Edge.** Transfer learning from graph classification tasks to edge prediction tasks within unrelated knowledge domains.

### A.4.1 NODE2GRAPH TRANSFER TASK

In the Node2Graph transfer scenario, we adapted the settings from the Graph2Node setup: during pre-training, we used the node-level ogbn-arxiv dataset to pre-train the model, while employing the graph-level downstream datasets used in the Graph2Graph tasks for fine-tuning. Moreover, we applied the same setup as in the Graph2Node scenario: The performance of GraphBridge was eval-

uated using a GIN backbone pre-trained with both the GraphCL and SimGRACE methods against fine-tuning, MetaFP and AdapterGNN. The experimental results are presented in Figure 20.

The experimental results of Node2Graph transfer show that our proposed GMST method dominates in this domain gap-obvious condition, both compared to the normal fine-tuning method and to the previous efficient tuning methods. Therefore, the additional experiments further reinforced the rules we previously established.

Table 20: **Results of Node2Graph Supplementary Transfer Scenario.** : Test ROC-AUC (%) performances on molecular prediction benchmarks with different workflows.

| Pre-train Methods | Tuning Methods | BACE | BBBP | ClinTox | HIV | SIDER | Tox21 | MUV | ToxCast | Avg. | Imp. |
|---|---|---|---|---|---|---|---|---|---|---|---|
| | FT | $72.3_{\pm1.7}$ | $67.1_{\pm2.5}$ | $68.2_{\pm2.7}$ | $75.4_{\pm2.4}$ | $58.9_{\pm1.9}$ | $72.3_{\pm1.5}$ | $73.0_{\pm3.4}$ | $61.7_{\pm1.4}$ | 68.7 | – |
| | MetaFP | $70.2_{\pm1.6}$ | $63.4_{\pm1.9}$ | $66.4_{\pm2.2}$ | $71.6_{\pm1.4}$ | $57.2_{\pm1.8}$ | $71.4_{\pm1.3}$ | $70.5_{\pm2.6}$ | $59.3_{\pm1.7}$ | 66.3 | -2.4% |
| GraphCL | Adapter | $70.6_{\pm2.1}$ | $64.0_{\pm1.4}$ | $65.2_{\pm1.8}$ | $71.7_{\pm1.9}$ | $56.7_{\pm2.3}$ | $71.5_{\pm1.5}$ | $72.0_{\pm2.1}$ | $60.1_{\pm2.3}$ | 66.5 | -2.2% |
| | **GSST** | $76.3_{\pm1.1}$ | $69.3_{\pm1.3}$ | $71.2_{\pm0.9}$ | $69.9_{\pm2.4}$ | $60.7_{\pm1.1}$ | $71.7_{\pm1.4}$ | $77.4_{\pm1.7}$ | $62.9_{\pm1.7}$ | 69.9 | 1.2% |
| | **GSMT** | $78.5_{\pm1.2}$ | $70.2_{\pm1.8}$ | $71.2_{\pm1.4}$ | $72.0_{\pm0.9}$ | $60.9_{\pm1.1}$ | $71.1_{\pm1.3}$ | $78.3_{\pm0.7}$ | $61.9_{\pm2.0}$ | **70.5** | 1.8% |
| | FT | $73.7_{\pm1.2}$ | $63.1_{\pm1.7}$ | $52.4_{\pm1.8}$ | $72.1_{\pm2.2}$ | $59.1_{\pm2.3}$ | $69.5_{\pm1.3}$ | $69.2_{\pm1.4}$ | $60.8_{\pm1.5}$ | 65.0 | – |
| | MetaFP | $71.2_{\pm2.3}$ | $60.0_{\pm1.9}$ | $50.1_{\pm2.3}$ | $69.1_{\pm1.8}$ | $56.6_{\pm2.5}$ | $68.9_{\pm1.5}$ | $69.1_{\pm1.9}$ | $59.2_{\pm2.3}$ | 63.3 | -1.7% |
| SimGRACE | Adapter | $71.9_{\pm1.7}$ | $60.6_{\pm1.8}$ | $50.9_{\pm2.2}$ | $70.3_{\pm1.7}$ | $55.2_{\pm1.9}$ | $69.4_{\pm1.6}$ | $70.4_{\pm2.4}$ | $59.3_{\pm2.0}$ | 63.5 | -1.9% |
| | **GSST** | $73.5_{\pm1.2}$ | $62.3_{\pm1.4}$ | $53.8_{\pm1.5}$ | $72.5_{\pm1.9}$ | $60.3_{\pm2.1}$ | $70.5_{\pm2.1}$ | $70.0_{\pm1.5}$ | $59.9_{\pm1.8}$ | 65.4 | 0.6% |
| | **GSMT** | $73.0_{\pm0.6}$ | $63.8_{\pm2.1}$ | $55.2_{\pm1.3}$ | $70.6_{\pm1.1}$ | $61.9_{\pm2.2}$ | $72.0_{\pm2.2}$ | $72.8_{\pm1.5}$ | $61.1_{\pm2.1}$ | **66.3** | 1.3% |

Table 21: **Results of Graph2Edge Supplementary Transfer Scenario.** : Test ROC-AUC (%) performances on diverse edge prediction benchmarks with different workflows.

| Pre-train Methods | Tuning Methods | Citeseer | | | PubMed | | | Cora | | | Amazon | | | Flickr | | |
|---|---|---|---|---|---|---|---|---|---|---|---|---|---|---|---|---|
| | | GCN | GAT | GIN | GCN | GAT | GIN | GCN | GAT | GIN | GCN | GAT | GIN | GCN | GAT | GIN |
| — | Scratch Train | $89.8_{\pm0.3}$ | $88.8_{\pm1.0}$ | $78.6_{\pm1.0}$ | $69.4_{\pm0.9}$ | $86.3_{\pm1.2}$ | $79.6_{\pm1.5}$ | $88.3_{\pm0.7}$ | $86.2_{\pm0.4}$ | $76.1_{\pm0.5}$ | $92.0_{\pm1.3}$ | $83.3_{\pm1.1}$ | $88.2_{\pm1.0}$ | $61.4_{\pm0.7}$ | $52.5_{\pm0.6}$ | $60.2_{\pm1.1}$ |
| | FT | $71.6_{\pm0.3}$ | $71.1_{\pm0.5}$ | $72.9_{\pm0.1}$ | $65.9_{\pm0.5}$ | $74.5_{\pm0.7}$ | $74.6_{\pm1.5}$ | $66.8_{\pm0.4}$ | $68.9_{\pm0.7}$ | $65.9_{\pm0.7}$ | $81.2_{\pm1.4}$ | $72.0_{\pm1.1}$ | $82.4_{\pm1.5}$ | $64.2_{\pm1.2}$ | $53.4_{\pm1.6}$ | $64.5_{\pm1.0}$ |
| | MetaFP | $67.8_{\pm0.3}$ | $68.7_{\pm1.1}$ | $66.3_{\pm1.2}$ | $60.2_{\pm0.7}$ | $66.1_{\pm1.5}$ | $64.3_{\pm1.0}$ | $62.3_{\pm1.1}$ | $62.0_{\pm0.8}$ | $61.5_{\pm0.4}$ | $72.5_{\pm1.1}$ | $74.1_{\pm1.3}$ | $74.5_{\pm1.3}$ | $55.8_{\pm1.6}$ | $48.8_{\pm0.9}$ | $52.2_{\pm1.3}$ |
| GraphCL | Adapter | - | - | $68.3_{\pm1.0}$ | - | - | $65.8_{\pm1.2}$ | - | - | $59.7_{\pm0.9}$ | - | - | $77.6_{\pm1.4}$ | - | - | $55.4_{\pm1.1}$ |
| | **GSST** | $68.6_{\pm0.4}$ | $75.2_{\pm0.4}$ | $72.2_{\pm0.5}$ | $58.6_{\pm0.7}$ | $77.1_{\pm0.2}$ | $74.1_{\pm0.7}$ | $74.5_{\pm0.7}$ | $74.6_{\pm1.5}$ | $64.9_{\pm0.6}$ | $80.2_{\pm1.3}$ | $80.0_{\pm1.6}$ | $82.2_{\pm1.3}$ | $60.4_{\pm1.0}$ | $51.2_{\pm1.6}$ | $61.7_{\pm1.2}$ |
| | **GMST** | $79.8_{\pm1.5}$ | $85.6_{\pm0.4}$ | $75.7_{\pm0.6}$ | $65.5_{\pm0.9}$ | $84.4_{\pm1.5}$ | $85.0_{\pm1.2}$ | $75.3_{\pm0.7}$ | $80.9_{\pm0.3}$ | $65.2_{\pm0.7}$ | $89.7_{\pm0.9}$ | $83.7_{\pm1.4}$ | $83.2_{\pm1.0}$ | $65.8_{\pm1.5}$ | $53.6_{\pm1.4}$ | $64.7_{\pm0.8}$ |
| | FT | $72.4_{\pm0.7}$ | $70.7_{\pm0.7}$ | $71.3_{\pm1.1}$ | $64.8_{\pm0.5}$ | $73.4_{\pm1.3}$ | $72.8_{\pm0.3}$ | $70.8_{\pm0.7}$ | $71.3_{\pm0.5}$ | $62.6_{\pm0.4}$ | $80.1_{\pm1.0}$ | $75.0_{\pm0.3}$ | $81.2_{\pm0.7}$ | $61.7_{\pm1.1}$ | $52.5_{\pm1.2}$ | $61.7_{\pm1.6}$ |
| | MetaFP | $64.1_{\pm0.9}$ | $65.7_{\pm1.0}$ | $65.1_{\pm1.1}$ | $58.3_{\pm1.2}$ | $63.2_{\pm0.1}$ | $65.4_{\pm1.1}$ | $61.3_{\pm0.7}$ | $60.5_{\pm0.8}$ | $60.4_{\pm1.0}$ | $70.3_{\pm0.1}$ | $73.2_{\pm0.7}$ | $73.8_{\pm1.3}$ | $53.8_{\pm1.4}$ | $48.4_{\pm1.0}$ | $53.3_{\pm1.8}$ |
| SimGRACE | Adapter | - | - | $61.3_{\pm1.3}$ | - | - | $65.0_{\pm0.8}$ | - | - | $60.8_{\pm1.0}$ | - | - | $74.6_{\pm0.9}$ | - | - | $53.2_{\pm1.1}$ |
| | **GSST** | $74.4_{\pm1.1}$ | $75.8_{\pm1.5}$ | $70.8_{\pm1.2}$ | $58.8_{\pm1.3}$ | $75.6_{\pm1.0}$ | $73.2_{\pm1.3}$ | $72.5_{\pm1.2}$ | $71.7_{\pm0.3}$ | $63.7_{\pm1.2}$ | $78.5_{\pm0.8}$ | $79.7_{\pm0.2}$ | $80.3_{\pm1.5}$ | $59.8_{\pm1.1}$ | $50.3_{\pm1.2}$ | $60.5_{\pm1.1}$ |
| | **GMST** | $75.2_{\pm1.1}$ | $80.6_{\pm1.5}$ | $74.7_{\pm0.7}$ | $65.0_{\pm1.7}$ | $80.8_{\pm1.5}$ | $81.7_{\pm1.3}$ | $74.4_{\pm0.6}$ | $77.9_{\pm1.6}$ | $65.2_{\pm0.9}$ | $85.4_{\pm0.6}$ | $80.2_{\pm1.1}$ | $82.0_{\pm1.3}$ | $63.3_{\pm1.4}$ | $52.5_{\pm1.5}$ | $62.0_{\pm1.1}$ |

### A.4.2 GRAPH2EDGE TRANSFER TASK

For the Graph2Edge transfer scenario, we used the same datasets, pre-training methods, and GNN backbones as in the Graph2Node setup. However, we reformulated the original node classification downstream task into an edge prediction task by sampling edges with both positive and negative examples. As the edge prediction task is a binary classification problem, we evaluated the performance using ROC-AUC scores. The results of this experiment are shown in Figure 21.

The results exhibited in Figure 21 indicate that our method achieves outstanding performance in the Graph2Edge transfer task. On the CiteSeer, PubMed, and Cora datasets, GMST consistently outperforms all fine-tuning methods, maintaining a clear competitive advantage. Moreover, our GMST breakthrough outperforms the fine-tuning method on the Amazon and Flickr considering Graph2Edge transfer.

In conclusion, by taking all the results from complementary scenarios into consideration, we find that GraphBridge demonstrates reliable performance across a variety of cross-domain transfer tasks, regardless of transfer task's complexity. This confirms the robustness and generalization capabilities of the GraphBridge framework, establishing it as an efficient and high-performing approach for graph transfer learning.

A.5   CONFUSION MATRIX VISUALIZATION OF GMST TUNING RESULTS

In order to display a clear visualization of the class-wise performance of the GMST method across different datasets, we take the Hard Task Scenario as an example to plot the confusion matrices for GMST results of various GNN backbones pre-trained with GraphCL on diverse node-classification benchmarks.

The results demonstrate that on CiteSeer, PubMed, and Cora datasets where GMST performs well, the predictions of the fine-tuned model are concentrated along the diagonal, indicating high accuracy across all categories. In contrast, for the Flickr and Amazon datasets, the fine-tuned model tends to predict test data as belonging to the category with the highest proportion in the training set, reflecting the impact of label imbalance during fine-tuning. These findings highlight that category imbalance in downstream task datasets can negatively influence the performance of the GMST algorithm.

Figure 5: **Visualization Results of GMST Fine-tuning in Hard Task Scenario on Different Datasets.** The visualized confusion matrices of all fine-tuned GNN backbones across five downstream task datasets are displayed.

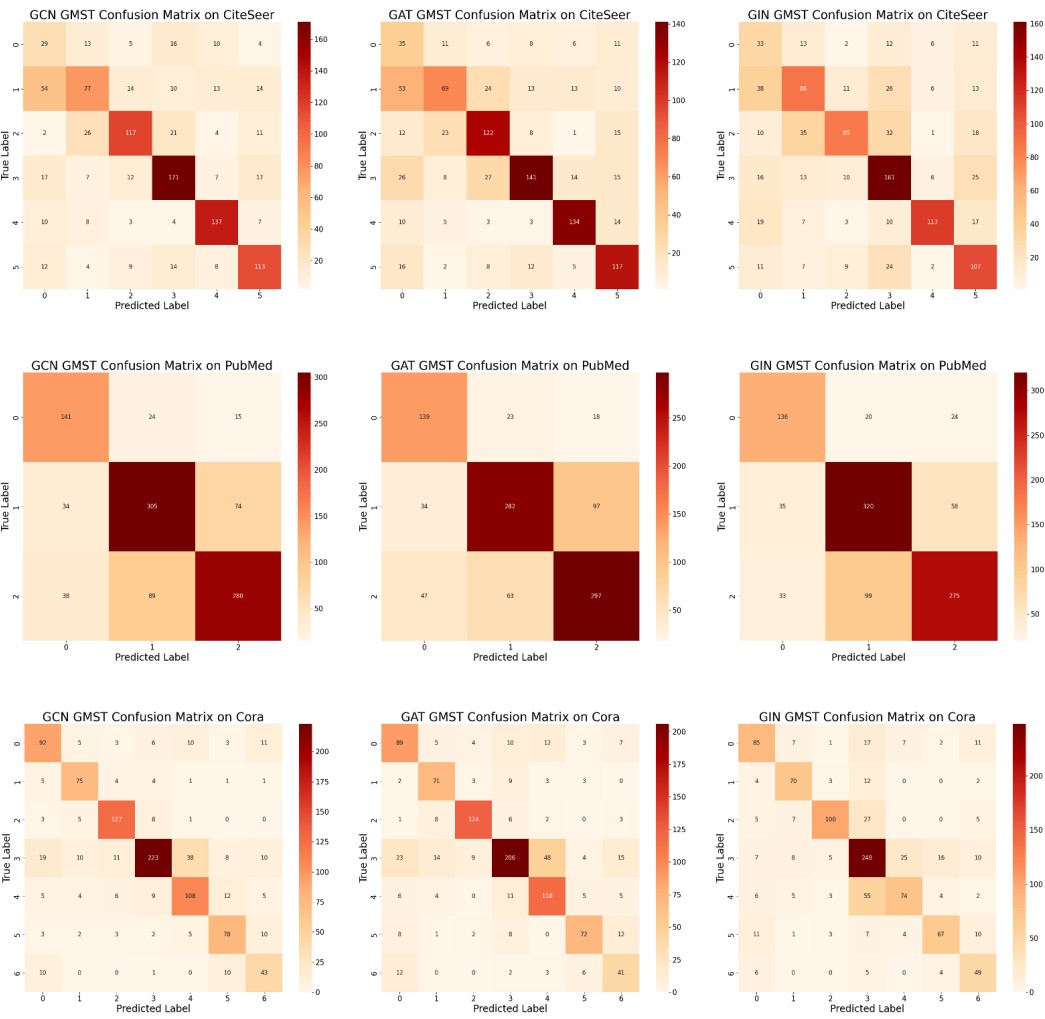

Figure 6: **Visualization Results of GMST Fine-tuning in Hard Task Scenario on Different Datasets. (Cont'd)**

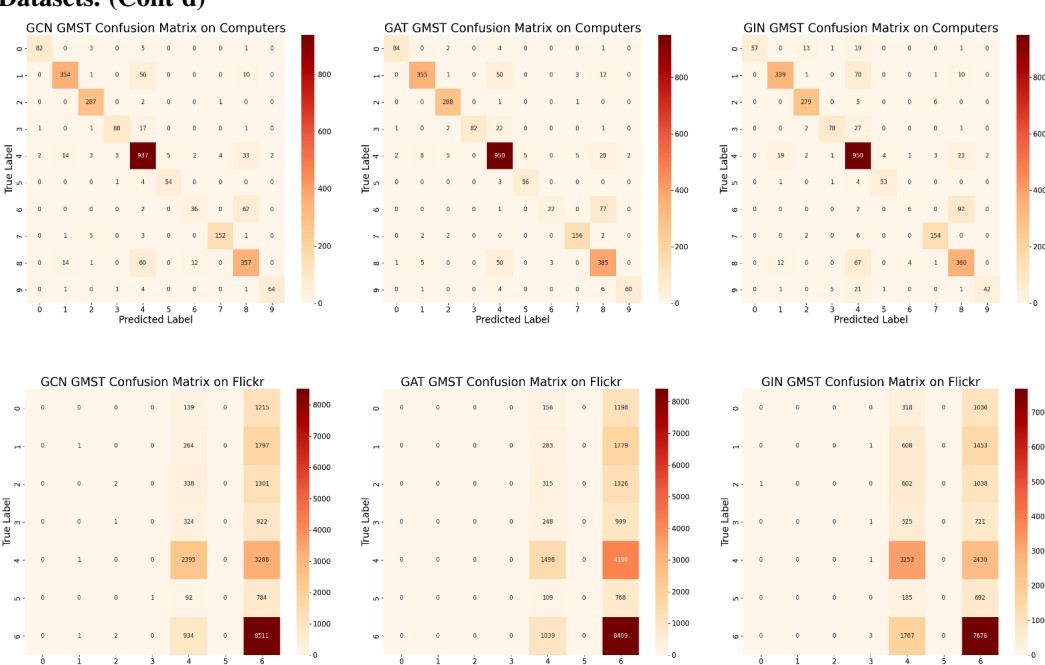

