# OpenReview forum: "GraphBridge: Towards Arbitrary Transfer Learning in GNNs"
_ICLR.cc/2025/Conference — ICLR 2025 Poster_

### Official Review · Reviewer_TMCF · 2024-10-31

**Soundness:** 3
**Presentation:** 3
**Contribution:** 3
**Rating:** 6
**Confidence:** 4

**Summary:**

In this scholarly article, the researchers introduce a novel framework termed GraphBridge, which is designed to facilitate resource-efficient graph transfer learning for arbitrary downstream tasks and domains. The overarching objective is to establish a unified workflow that maximizes the utility of pre-trained Graph Neural Networks (GNNs) across a spectrum of cross-level and cross-domain downstream tasks, thereby eliminating the necessity for data reorganization and task reformulation. To achieve this, the authors have constructed four distinct scenarios of graph transfer learning tasks, spanning from simple to complex, and proposed two resource-efficient tuning methods, namely Graph-Scaffold Side-Tune (GSST) and Graph-Merge Side-Tune (GMST), to address the associated challenges. The experimental validation, conducted across diverse domains and tasks, including graph and node classification as well as 3D object recognition, demonstrates the effectiveness of their approach in enabling arbitrary domain transfer learning on GNNs with enhanced resource efficiency.

**Strengths:**

1. The paper introduces the GraphBridge framework, an innovative solution aimed at enabling arbitrary transfer learning of Graph Neural Networks (GNNs) across different tasks and domains, an area that has not been fully explored in previous research. GraphBridge, through the introduction of prediction heads and a bridging network, allows pre-trained GNNs to effectively transfer learning between inputs and outputs of different dimensions, a novel design approach.
2. The paper has conducted extensive experimental validation on multiple datasets, covering a range of transfer learning scenarios from simple to complex, demonstrating the rigor and reliability of the research methodology.
3. The paper is structured clearly, with a tight logic, detailing the problem statement, methodology, experimental design, and result analysis, making it easy for readers to understand and follow.
4. The paper addresses an important issue in the practical application of GNNs, that is, how to effectively transfer and reuse knowledge trained on one type of data to another different type of data, which is of great significance for promoting the widespread application of GNNs.

**Weaknesses:**

1. In the research of graph neural networks, the interpretability of models is an important aspect. The paper may need to further explore how to explain the decision-making process of GraphBridge.
2. The paper may need to provide more visualizations of the experimental results, such as confusion matrices, to help readers more intuitively understand the model's performance.

**Questions:**

In the realm of graph neural networks research, the interpretability of models is a crucial consideration. The paper might need to delve deeper into elucidating the decision-making mechanisms of GraphBridge. Furthermore, to aid readers in gaining a more intuitive grasp of the model's performance, the paper could benefit from incorporating additional visual representations of the experimental outcomes, such as confusion matrices. In the HARD TASK challenge, as shown in Table 3, the model's performance on the Amazon and Flickr datasets does not match the effectiveness seen with FT. Please provide an explanation considering aspects such as the structure of the datasets.

---

> ### Author Response · Authors · 2024-11-20
> **Answers to Q1-Q3**
>
> `>>> Q1` **Interpretability of GraphBridge decision-making**
>
> `>>> A1` Thanks for the reviewer’s comment. To explain the decision-making process of GraphBridge, we explored the **adaptive variation of the learnable alphas between the primary and side networks under different task scenarios**. Using the GMST tuning of the GCN backbone on the Cora dataset as an example, we compared the final alphas after tuning between the primary and side networks under two scenarios: Node2Node and Graph2Node, to illustrate how GraphBridge determines the proportion of primary and side network fusion under different difficulties of transfer scenarios.
>
> First, we analyzed the layer-wise similarity between the parameters of the primary and side networks using the **CKA-linear** method proposed in [1], to confirm that the primary and side networks capture distinct encoding patterns across different task domains (primary network for pre-training domain, side network for downstream domain). The results show a similarity of 0.33 in layer 1 and 0.27 in layer 2 as expected, indicating the divergence in their feature embedding. Then, we compared the final alphas for the primary-side network merge in both Node2Node and Graph2Node scenarios:
>
> | | Node2Node | Graph2Node |
> | --- | --- | --- |
> | **Layer.0 alpha** | 0.412 |  0.293 |
> | **Layer.1 alpha** | 0.339 |  0.215 |
>
> The findings reveal that the layer-wise alphas in the Node2Node scenario are significantly higher than those in the Graph2Node scenario. This suggests that when the upstream and downstream task domains’ gap is narrower, more knowledge from the pre-trained model is transferable to the downstream task. Consequently, GMST adaptively fuses a larger proportion of embeddings from the pre-trained model into the side network via the oriented adjustment of alpha. Conversely, for the Graph2Node task with greater inter-domain differences, the fusion proportion from the pre-trained model is lower. These results demonstrate the interpretability of our approach, as GMST adaptively learns the optimal fusion ratio between the primary and side networks to accommodate domain differences.
>
> [1] Similarity of Neural Network Representations Revisited
>
> `>>> Q2` **Lack of visualization results**
>
> `>>> A2` Thanks for the advice. To provide a clear visualization of the class-wise performance of the GMST method across different datasets, we have plotted the confusion matrices for GMST results using various GNN backbones pre-trained with GraphCL on Node Classification datasets under the hard task Scenario. These visualizations have been included in Section A5 of the Appendix. We invite you to refer to the revised paper for the detailed results.
>
> `>>> Q3` **Marginal performance gain on Amazon and Flickr**
>
> `>>> A3` We appreciate the reviewer’s comment. In the middle and hard task scenarios, GMST does not outperform FT on the Amazon and Flickr datasets, primarily due to the imbalance in label distribution within these datasets. The label distributions for Amazon and Flickr are shown below, revealing that certain categories in both datasets have significantly more samples than others:
>
> - Amazon — Class0: 436, Class1: 2142, Class2: 1414, Class3: 542, **Class4: 5158**, Class5: 308, Class6: 487, Class7: 818, Class8: 2156, Class9: 291
> - Flickr — Class0: 5264, Class1: 8506, Class2: 6413, Class3: 4903, **Class4: 22966**, Class5: 3479. **6: 37719**
>
> This imbalance creates a substantial bias in the model fine-tuning. As a PEFT method, GMST is slightly less robust to such bias compared to the FT method and, therefore fails to outperform it. In contrast, for the CiteSeer, PubMed, and Cora datasets, the training sets used for fine-tuning are balanced, as they follow the official data split with an equal number of samples per category, enabling GMST to effectively outperform the FT method in the final fine-tuning results.

---

### Official Review · Reviewer_TZwq · 2024-11-03

**Soundness:** 3
**Presentation:** 4
**Contribution:** 4
**Rating:** 8
**Confidence:** 3

**Summary:**

In this paper, a framework is introduced, which aims to realize the arbitrary transfer learning of graph neural network (GNN) between different tasks and fields.Demonstration that GraphBridge achieves high efficiency with significantly fewer parameters (5%-20% of typical models) while maintaining comparable performance.

GraphBridge incorporates Graph Side-tuning techniques—Graph-Scaff-Side-Tune (GSST) and Graph-Merge-Side-Tune (GMST)—to minimize negative transfer effects, which occur when knowledge transfer from one domain adversely impacts performance in another. GSST focuses on efficiently transferring knowledge using a side network for layer-wise fusion with the base model. GMST, designed for domains with larger gaps, merges a pretrained model with a randomly initialized model, effectively reducing bias from the source domain.

At the same time, the author created a "task pyramid" with four levels of difficulty and tested it on 16 data sets.

**Strengths:**

Originality:GraphBridge introduces a novel approach to arbitrary transfer learning within Graph Neural Networks (GNNs), allowing models to be reused across a variety of tasks and domains without modifying task-specific configurations or graph structures. This approach is particularly original in that it combines prediction heads and a bridging network in a way that facilitates diverse task adaptation while preserving pretrained model knowledge.

Quality:The paper presenting a comprehensive evaluation of GraphBridge across 16 datasets and multiple transfer scenarios, ranging from relatively straightforward graph-to-graph tasks to more complex cross-domain transfers.

Clarity:The paper is generally clear in its exposition of both the problem and the proposed solution and the diagram of the paper is very clear and beautiful, which is pleasing to the eye.

Significance:GraphBridge is significant because it offers a flexible, task-agnostic framework for GNN transfer learning—a major advance in the field. The framework has broad implications for practical applications across various domains where graph data is prevalent, from molecular property prediction to social networks.

**Weaknesses:**

While the paper introduces two side-tuning techniques (GSST and GMST) to address negative transfer, it would be beneficial to delve deeper into when and why each approach is likely to succeed or fail. For instance, providing clearer conditions under which GSST vs. GMST is preferable would help practitioners select the appropriate method for specific tasks. Additionally, it might be useful to add a comparison with other common techniques for mitigating negative transfer, such as domain adaptation approaches, as this would position the side-tuning techniques more clearly within the broader context of transfer learning.

**Questions:**

1.How to prove the wide applicability of the task pyramid?
2.Can you provide more specific guidance on when to use GSST versus GMST?

---

> ### Author Response · Authors · 2024-11-20
> **Answers to Q1-Q3**
>
> `>>> Q1` **Proof of the wide applicability of the task pyramid**
>
> `>>> A1` Thanks for your comment. Our proposed task pyramid organizes the most common and pressing scenarios in graph transfer learning—Graph2Graph, Node2Node, and Graph2Node—based on the complexity of the task implementation, and even explores the transfer from static graph data to point cloud graph data. To date, our task pyramid encompasses the most comprehensive range of graph transfer scenarios. Additionally, beyond the transfer scenarios discussed in the main text, we also conduct experiments on Graph2Edge and Node2Graph scenarios, as detailed in **Appendix A4**. These results further demonstrate the broad applicability and flexibility of our GraphBridge framework. Overall, the GraphBridge addresses transfer learning challenges across all primary graph task levels (Graph-level, Node-level, and Edge-level), enabling effective transfer learning between arbitrary two task domains.
>
> `>>> Q2` **Guidance on when to use GSST versus GMST**
>
> `>>> A2` We apologize for not giving a clear and concrete principle to guide the selection of algorithms from GSST and GMST under different task scenarios. We provide some insight into the methods selection in **lines 456–459 of the main text**. To clarify the applicable scenarios of GSST, we make a further summary as follows:
>
> GSST is specifically suitable for scenarios where both the pre-training task and the downstream task are graph-level tasks, e.g. Graph2Graph (Table 1) and Graph-level datasets to Point Cloud datasets task (Table. 4, left side). This is because GSST lacks effective modules for mitigating negative transfer. As a result, it can only transfer pre-trained knowledge without being impacted by noise when the pre-training method, pre-training task, and downstream task operate at the same task level. Conversely, GMST is more effective when there is a significant gap between source and target domains, e.g. Node2Node (Table 2), Graph2Node (Table 3), and Node-level datasets to Point Cloud datasets task (Table 4, right side). In those scenarios, GSST only serves as the comparison method to demonstrate the capability of GMST in mitigating negative transfer when a large domain gap exists.
>
> `>>> Q3` **Lack of domain adaptation methods for comparison**
>
> `>>> A3` Thanks for the advice. To be honest, domain adaptation methods are typically limited to transferring knowledge between similar domains at the same task level, making them unsuitable for a fair comparison with our framework, which is designed for arbitrary domain transfer tasks. To address your concerns, we conduct relatively fair comparisons with **the results of classic domain adaptation algorithms AdaGCN and UDAGCN, alongside the results of our GMST on GCN under the Node2Node transfer scenario**, and record them in the table below:
>
> | Methods (Acc %) | Citeseer | PubMed | Cora | Amazon | Flickr |
> | --- | --- | --- | --- | --- | --- |
> | AdaGCN | 53.60 | 68.80 | 68.00 | 70.16 | 38.89 |
> | UDAGCN | 55.80 | 70.10 | 68.70 | 72.32 | 37.46 |
> | GraphCL pre-trained GCN + GMST | 59.30 | 72.10 | 73.10 | 89.42 | 51.92 |
> | SimGRACE pre-trained GCN + GMST | 61.60 | 73.20 | 75.10 | 90.88 | 50.56 |

---

### Official Review · Reviewer_i5HC · 2024-11-04

**Soundness:** 3
**Presentation:** 2
**Contribution:** 2
**Rating:** 6
**Confidence:** 4

**Summary:**

This paper proposes GraphBridge, a framework for knowledge transfer across different tasks and domains in Graph Neural Networks (GNNs). The main contributions include 1) An end-to-end pre-training-tuning framework, 2) Two parameter-efficient transfer methods (GSST and GMST) based on side-tuning, 3) A "Task Pyramid" defining different difficulty levels and validation on 16 datasets.

**Strengths:**

- The paper addresses the challenging issue of cross-task and cross-domain transfer in GNNs.
- The design of using side-tuning to mitigate negative transfer is rational.
- The experiments are conducted across multiple scenarios and datasets to evaluate the performance.

**Weaknesses:**

- The technical novelty seems limited. The overall framework is a straightforward application of traditional pre-train finetune paradigm but lacks graph-specific innovative designs.
- The paper directly uses MLP as a side network without comparing it to other networks, such as co-attention used by multimodal large language models in bridging two different modalities.
- The improvements seem quite limited in some datasets, e.g., some results in Table 1. Can the authors elaborate more on the possible reasons?
- The analysis of computational efficiency and memory consumption is not sufficient. It would be better to conduct a comparison from a theoretical or experimental perspective for further discussion.
- In Loss, alpha is learnable, but there seems to be no mechanism to limit the range of alpha. Will this cause alpha of gnn to be 0 and the mlp part to dominate?
- What are the criteria for choosing between GSST and GMST?

**Questions:**

See weaknesses.

---

> ### Author Response · Authors · 2024-11-20
> **Answers to Q1-Q3**
>
> `>>> Q1` **Limited novelty in technique**
>
> `>>> A1` Thanks for the comment. We need to recognize that graph-specific modular innovations are not highlighted in our tuning architecture design. However, the GraphBridge framework represents a significant innovation in the field of graph transfer learning, since it is the first generic framework capable of tackling the challenging task of arbitrary graph domain transfer learning, which incorporates various novel designs specifically aimed at bridging the task and knowledge gap unique to graph transfer.
>
> Furthermore, as for the tuning architecture design, we employed the MLP as the side network, informed by our previous research on the interchangeability of MLP and GNN models[1] [2], enhancing efficiency while maintaining competitive performance. This design can only work under the circumstance of **graph transfer learning**. Additionally, we introduced replaceable task heads in our tuning model to address the challenges of cross-task domain transfer that is specific to **graph data**.
>
> [1] Mlpinit: Embarrassingly simple gnn training acceleration with mlp initialization
>
> [2] Graph-less neural networks: Teaching old  mlps new tricks via distillation
>
> `>>> Q2` **Lack of ablation study on side network structure**
>
> `>>> A2` We apologize for not including ablation experiments on the side network structure in our paper. However, the co-attention structure used in multimodal models is not fit for our Graph Side-Tuning methods, since the main and side networks in our framework are alternately stacked to merge the hidden layer outputs, and thus cannot compute the attention of the two branches at the same layer. To address your concerns, we tried **using each GNN backbone's corresponding lightweight structure as a side network, experimented on Middle and Hard task settings**, and extensively compared the results with those of the MLP side network. The detailed results of these experiments are recorded in Section A3.4 of the Appendix in the revised paper, and we invite you to refer to it for the complete ablation experiment results and corresponding analysis.
>
> `>>> Q3` **Marginal performance gain on some dataset**
>
> `>>> A3` Thanks for the reviewer's comment. The limited performance improvement of our GSST on certain datasets in the Easy Scenario (Table 1) is attributed to the following 2 reasons:
>
> 1. **Limitations of pre-training methods:** From the experimental results, it can be observed that the models pre-trained using GraphCL and fine-tuned with GSST surpass the performance of FT on more downstream datasets compared to those pre-trained using SimGRACE method. This indicates that as the quality of pre-training improves, it equips the model with more generalizable pre-training knowledge, thereby enhancing the domain adaptation capability of PEFT methods for downstream tasks. However, as current graph pre-training methods still cannot achieve highly generalized ‘world knowledge’ as LLM does, the performance degradation on some data when using PEFT methods is inevitable.
> 2. **Imbalanced label distribution in datasets:** An analysis of the label distributions in datasets where GSST showed limited performance improvement reveals severe imbalances. For instance, the ratio of positive to negative samples in the **HIV** and **Tox21** datasets is as high as **27:1** and **22:1**, respectively. This extreme imbalance results in significant bias during model tuning, limiting the effectiveness of GSST. As a PEFT method, GSST is less robust to such label imbalance compared to the FT method.

---

> ### Author Response · Authors · 2024-11-20
> **Answers to Q4-Q6**
>
> `>>> Q4` **Insufficient analysis of efficiency**
>
> `>>> A4` We appreciate the reviewer’s comment. In fact, we have demonstrated the parameter efficiency and fine-tuning speed-up of GSST and GMST from an experimental perspective, as discussed in **lines 485-521 of Section 4.6** in the main text. For more detailed results, please refer to **Figure 3** and **Table 6**. Besides, we apologize for not including the theoretical discussion of the tuning efficiency. In response, we compare the time complexity of our PEFT method and the full fine-tune method on the GCN backbone to display the tuning efficiency of our approach theoretically.
>
> **For the FT method:** Assuming the graph consists of $n$ nodes and $m$ edges, with $k$ representing the number of the convolutional layers in GCN and $d$ denoting the feature dimension of each node, then the time complexity of the GCN can be represented as $O(knm^2+knd^2)$. This expression contains 2 components: firstly, $O(knm^2)$ accounts for the computation of the m elements of the adjacency matrix at each graph convolutional layer, while the second component, $O(knd^2)$, reflects the matrix computation involving the node features. In the contexts of forward propagation and back-propagation, the time complexities of the model are similar; thus, the time complexity of the GCN during the standard fine-tuning stage is $O(2knm^2+2knd^2)$
>
> **For the proposed PEFT method:** The side network of our PEFT method is constructed using MLP architecture, resulting in a time complexity of $O(knd^2)$ for processing the same graph data. Furthermore, since the forward propagation processes of both the backbone network and the side network occur in parallel, the overall time complexity during forward is constrained by the GCN backbone. Therefore, the time complexity for the forward propagation is given by $O(knm^2+knd^2)$. However, the time complexity reduces to $O(knd^2)$ during back-propagation since the GCN backbone incurs no computational cost in our side-tuning architecture. In summary, the total time complexity of the PEFT model can be expressed as $O(knm^2+2knd^2)$, demonstrating that the proposed PEFT method enhances efficiency by eliminating the need for back gradient propagation on the GNN during the tuning stage.
>
> `>>> Q5` **The values of alphas at convergence**
>
> `>>> A5` Thank you for your question. To address your concerns, we use the pre-trained GCN backbone as an example to present the layer-wise learning results of the alphas between the pre-trained and side networks after GMST tuning on different datasets under the Hard Task Scenarios and the results are shown below:
>
> | 2-layer GCN | CiteSeer| PubMed| Cora| Amazon| Flickr|
> | --- | --- | --- | --- | --- | --- |
> | **Layer.0 alpha** | 0.257 | 0.326 | 0.339 | 0.309 | 0.185 |
> | **Layer.1 alpha** | 0.199 | 0.199 | 0.215 | 0.365 | 0.379 |
>
> The results indicate that the alpha at each layer does not converge to the extreme case of 0, even without any constraints. Notably, all alpha values are initialized to 0 but ultimately converge to a range of 0.15 to 0.4 after tuning. It demonstrates that our fine-tuning algorithm adaptively extracts knowledge beneficial to downstream tasks from the primary network based on the generalization strength of the pre-training domain to the downstream domain, achieving optimal fine-tuning performance through this adaptive merge.
>
> `>>> Q6` **Criteria of tuning methods selection**
>
> `>>> A6` We apologize for not giving a clear and concrete criteria to guide the selection of algorithms from GSST and GMST under different task scenarios. In fact, we provide some insight of the tuning methods selection in **lines 456–459 of the main text**. To further clarify the criteria, we summarize it as follows:
>
> **Criteria:** GSST is only suitable for scenarios where the domain gap between the pre-training task and the downstream task is minimal, and the pre-training method, pre-training task, and downstream task implement at the same level (Graph-level), e.g. Graph2Graph (Table 1) and Graph-level datasets to Point Cloud datasets task (Table. 4, left side); Conversely, GMST is more effective when there is a significant gap between source and target domains, e.g. Node2Node (Table 2), Graph2Node (Table 3), and Node-level datasets to Point Cloud datasets task (Table 4, right side).

---

### Official Review · Reviewer_G82U · 2024-11-04

**Soundness:** 4
**Presentation:** 2
**Contribution:** 4
**Rating:** 8
**Confidence:** 4

**Summary:**

In this paper, a novel GraphBridge framework is proposed. This framework introduces a resource-efficient approach for enabling versatile graph transfer learning across diverse tasks and domains, designed to fully leverage pre-trained Graph Neural Networks (GNNs) without necessitating data restructuring or task redefinition. It outlines four transfer learning scenarios of varying complexity and employs two novel tuning techniques, GSST and GMST, to navigate specific challenges in each. Experimental results across multiple domains, covering tasks like graph and node classification as well as 3D object recognition, validate GraphBridge's effectiveness in achieving domain-agnostic transfer learning with greater resource efficiency. Future work will expand the framework's applicability by addressing more extensive benchmarks.

**Strengths:**

GraphBridge offers a versatile framework for efficient graph transfer learning, enabling pre-trained GNNs to tackle diverse tasks and domains without extensive reorganization. It achieves resource efficiency through two novel tuning methods, GSST and GMST. The framework supports both cross-level and cross-domain tasks, allowing scalable application across simple to complex scenarios. Extensive experiments demonstrate its adaptability and robustness.

**Weaknesses:**

Although the paper represents a good novel contribution, there are some issues, which are as follows:

1. The framework relies on high-quality pre-trained GNNs, which may not always be available or easy to obtain.
2. Some domain-specific nuances might still require additional adaptation, potentially limiting GraphBridge’s effectiveness in highly specialized applications.
3. Baselines are very less and not that recent may be due to the scope of the work.

**Questions:**

The work covers nearly all aspects which should be taken care for the study. However, there are following question which may arise:
1. Hypergraphs are also a way which considers information with their context for downstream tasks. Can the framework be adapted for hypergraphs as well or can it be comapred with hypergraphs?
2. How can it be compared statistically with the given baselines?

---

> ### Author Response · Authors · 2024-11-20
> **Answers to Q1-Q4**
>
> `>>> Q1` **Limitation caused by pre-trained model dependency**
>
> `>>> A1` Thanks for the reviewer's comment. We need to admit that while our framework offers a general solution for graph transfer learning across diverse task domains, it relies on effective pre-training of GNNs. Fortunately, with ongoing advancements in GNN pre-training techniques, the performance of GNN pre-training continues to improve. For instance, recent methods such as **GraphMAE**[1], **GraphLoG**[2], and **DGPM**[3] have demonstrated excellent results in graph-level transfer tasks. Given the flexibility of our framework, these pre-training methods can be effectively integrated with GraphBridge to enable more efficient transfer learning across arbitrary task domains.
>
> [1] GraphMAE: Self-Supervised Masked Graph Autoencoders
>
> [2] Self-supervised Graph-level Representation Learning with Local and Global Structure
>
> [3] Empowering Dual-Level Graph Self-Supervised Pretraining with Motif Discovery
>
> `>>> Q2` **Influence of domain-specific nuances towards GraphBridge’s effectiveness**
>
> `>>> A2` We appreciate the reviewer’s comment. To ensure consistency in the experiments, we have adopted uniform model structures and pre-training configurations across all task scenarios. However, as you pointed out, domain-specific knowledge in different graph datasets may necessitate specialized structures or methods to adapt. For example:
>
> 1. It is more common to employ DGCNN rather than GCN, GAT and GIN in point cloud classification tasks, to capture both local graph information and geometric semantics.
> 2. It is necessary to adapt the feature space between molecular and citation datasets when different approaches are implemented for node feature encoding.
>
> To address **issue 1**, we have implemented a tailored design within the input bridge component of our framework. As described in **lines 209–210 of the main text**, we project 3D point cloud features through a MLP projector to generate new representations compatible with structures including GCN, GAT and GIN for input into the pre-trained GNN backbone and train the projector simultaneously during the fine-tuning to ensure that domain-specific knowledge is harmonized as effectively as possible. Furthermore, we will also try to solve **issue 2** in our future work.
>
> `>>> Q3` **Lack of baselines for comparison**
>
> `>>> A3` Thanks for the comment. Due to the novelty of our task setup, identifying appropriate baseline methods for fair comparison is challenging. To alleviate your concerns, we incorporate the SOTA domain adaptation methods: **AdaGCN** and **UDAGCN** methods as baselines in the **Node2Node scenario** for a more comprehensive comparison. Since both AdaGCN and UDAGCN utilize the GCN as their backbone, we also use **the results of GMST implemented on the GCN** for further comparison. The results are presented in the following table:
>
> | Methods (Acc %) | Citeseer | PubMed | Cora | Amazon | Flickr |
> | --- | --- | --- | --- | --- | --- |
> | AdaGCN | 53.60 | 68.80 | 68.00 | 70.16 | 38.89 |
> | UDAGCN | 55.80 | 70.10 | 68.70 | 72.32 | 37.46 |
> | GraphCL pre-trained GCN + GMST | 59.30 | 72.10 | 73.10 | 89.42 | 51.92 |
> | SimGRACE pre-trained GCN + GMST | 61.60 | 73.20 | 75.10 | 90.88 | 50.56 |
>
> `>>> Q4` **Missing of discussion on hypergraph data**
>
> `>>> A4` Thank you for the question. To address it, we explored the application of the GraphBridge framework to a hypergraph node classification task. Specifically, we selected the **‘Mushroom’ hypergraph node classification dataset** and transformed the hypergraph data to common graph data using the **‘Line Expansion’ algorithm** proposed in [1]. This transformation allowed us to simplify the hypergraph node classification task into a standard node classification task. Therefore, we are taking the pre-trained GCN backbone to implement transfer learning from the original pre-training domain to the hypergraph task domain, following the Node2Node and Graph2Node task configurations. We then compared the performance of different fine-tuning methods, including Fine-tuning, GSST and GMST.
>
> | Scenario | Scratch Training | FT | GSST | GMST |
> | --- | --- | --- | --- | --- |
> | Node2Node | 96.2 | 96.5 | 96.4 | **96.9** |
> | Graph2Node | 96.2 | 93.6 | 93.4 | **96.8** |
>
> The results indicate that, with the appropriate hypergraph expansion strategy, our framework effectively transfers knowledge from the pre-trained domain to the hypergraph task domain. This demonstrates the adaptability of our approach and confirms its applicability to transfer tasks involving hypergraphs.
>
> [1] Semi-supervised Hypergraph Node Classification on Hypergraph Line Expansion

---

> > ### Author Response · Authors · 2024-11-20
> > **Answer to Q5**
> >
> > `>>> Q5` **Lack of statistical significance in experimental results**
> >
> > `>>> A5` Thanks for your comment. To address your concerns regarding the reliability of our experiments, we did the statistical analysis using Graph2Graph task transfer as an example. Specifically, we calculate the average P-values using the T-score for GSST against FT, MetaGP, MetaFP and AdapterGNN across different downstream datasets to determine the significance of the proposed tuning algorithm. The results are shown in the table below.
> >
> > | Comparison | Ave. P-value | Significance (Threshold=0.05) |
> > | --- | --- | --- |
> > | GSST v.s. FT | 0.0461 | Yes |
> > | GSST v.s. MetaGP | 0.0323 | Yes |
> > | GSST v.s. MetaFP | 0.0472 | Yes |
> > | GSST v.s. AdapterGNN | 0.0503 | No |
> >
> > The results indicate that GSST achieves statistically significant improvements compared to nearly all baseline methods, proving that the GSST algorithm is more effective than previous fine-tuning methods.

---

### Public Comment · ~Wei_Zhuo4 · 2025-09-22
**How to reproduce the reported results?**

Dear authors,

While reading the paper, I found several aspects of the proposed model unclear. For example, the model structure of **Feat-Adapt**. In addition, the codebase (https://github.com/jujulili888/GraphBridge) is not well organized and lacks a README to guide reproduction. Could you please complete the repository and provide detailed settings (environment, datasets, hyperparameters, and training scripts) to reproduce the reported results? This would be very valuable to the community.

Thanks

---

### Meta-Review · Area_Chair_wRer · 2024-12-20

**Metareview:**

**Summary:**

This paper proposes a new approach for transfer learning across diverse graph tasks and domains. The proposed framework introduces a resource-efficient approach that leverages pre-trained Graph Neural Networks (GNNs) without restructuring data or redefining tasks. The authors developed an end-to-end framework and proposed two parameter-efficient transfer learning methods (GSST and GMST) using side-tuning. Finally, Task Pyramid was presented to discuss difficulty levels.

**Strengths:**

1. **Significance.** This paper studies a relatively less explored problem, transfer learning for GNNs. The proposed method successfully adapts GNNs across a variety of graph tasks and domains without significant task or domain specific modifications.
2. **Evaluation on various datasets and settings.** This paper demonstrates the effectiveness of the proposed methods across 16 datasets with diverse transfer learning scenarios, including Graph2Graph, Node2Node, Graph2Node, and Graph2Point-Cloud.
3. **Presentation.** This paper is well-written and easy to follow.

**Weaknesses:**

1. **Weak baselines.** Although the proposed method was evaluated in various settings, a relatively small number of baselines are included. This limitation is partly due to the fact that this problem is underexplored.
2. **No systematic way to select an algorithm.** Two methods GSST and GMST have been proposed in the paper, but the authors did not provide any insights on how to choose between them.

**Main reasons:**

This paper is well-written and tackles an underexplored yet important problem. Transfer learning for GNNs is a challenging problem and it has the potential to accelerate  research on GNN foundation models. The proposed method requires minimal task/domain-specific modification and is evaluated across 16 benchmark datasets in various settings.

**Additional Comments On Reviewer Discussion:**

(1) Weak baselines and (2) computational efficiency were concerns raised by reviewers.
1. Since transfer learning for graph neural networks is underexplored, a relatively small number of baselines is understandable.
2. Additionally, the authors addressed computational efficiency by providing theoretical time complexity analyses.
Overall, the authors addressed most concerns during the rebuttal.

---

### Decision · Program_Chairs · 2025-01-22

Accept (Poster)